# CloDS: Visual-Only Unsupervised Cloth Dynamics Learning in Unknown Conditions

**Yuliang Zhan**[1,†]**, Jian Li**[1,†]**, Wenbing Huang**[1]**, Yang Liu**[2]**, Hao Sun**[1,*]
[1]Gaoling School of Artificial Intelligence, Renmin University of China, Beijing, China
[2]School of Engineering Science, University of Chinese Academy of Sciences, Beijing, China
Emails: zhanyuliang@ruc.edu.cn (Y.Z.); haosun@ruc.edu.cn (H.S.)

## Abstract

Deep learning has demonstrated remarkable capabilities in simulating complex dynamic systems. However, existing methods require known physical properties as supervision or inputs, limiting their applicability under unknown conditions. To explore this challenge, we introduce Cloth Dynamics Grounding (CDG), a novel scenario for unsupervised learning of cloth dynamics from multi-view visual observations. We further propose Cloth Dynamics Splatting (CloDS), an unsupervised dynamic learning framework designed for CDG. CloDS adopts a three-stage pipeline that first performs video-to-geometry grounding and then trains a dynamics model on the grounded meshes. To cope with large non-linear deformations and severe self-occlusions during grounding, we introduce a dual-position opacity modulation that supports bidirectional mapping between 2D observations and 3D geometry via mesh-based Gaussian splatting in video-to-geometry grounding stage. It jointly considers the absolute and relative position of Gaussian components. Comprehensive experimental evaluations demonstrate that CloDS effectively learns cloth dynamics from visual data while maintaining strong generalization capabilities for unseen configurations. Our code is available at https://github.com/whynot-zyl/CloDS. Visualization results are available at https://github.com/whynot-zyl/CloDS_video.

## 1 Introducion

Accurately modeling complex dynamical systems is fundamental to forecasting (Si & Chen, 2025; Li et al., 2025a), control (Wei et al., 2025; Liu et al., 2025), and optimization (Zhou et al., 2024; Tang et al., 2025) across diverse natural and engineered domains. Recently, deep learning has revolutionized the simulation of various dynamic systems, including fluid (Zeng et al., 2025), cloth (Li et al., 2025b), and multi-body dynamics (Carleo & Troyer, 2017). However, current approaches are highly dependent on supervision derived from physics and environmental properties in known conditions (*e.g.,*, particles, meshes) (Pfaff et al., 2020; Wang et al., 2024c). This dependency hinders the feasibility of unsupervised dynamics learning from visual data in robotics (Finn et al., 2016; Lee et al., 2018) and computer vision (Shi et al., 2015; Wang et al., 2017; Li et al., 2025c), where physical properties under an unknown environmental condition are inaccessible.

To address this challenge, intuitive physics approaches inspired by human perception have emerged. These approaches enable unsupervised dynamics learning directly from visual data (Wu et al.; Li et al., 2025a). Existing intuitive physics approaches have made notable progress in modeling rigid-body interactions (*e.g.,*, collisions and forces). However, they still fall short in deformable continuum mechanics, particularly cloth dynamics. This paper explores a novel scenario in intuitive physics: **C**loth **D**ynamics **G**rounding (**CDG**), aimed at unsupervised learning cloth dynamics from a series of multi-view videos. CDG is challenging due to the infinite-dimensional state

Table 1: Capabilities of different vision-only methods.

| Method | 3D Dynamic Learning | Video Prediction | Dynamic Scene Novel View Synthesis |
|---|---|---|---|
| 3D reconstruction | ✗ | ✗ | ✓ |
| Video prediction | ✗ | ✓ | ✗ |
| CloDS (ours) | ✓ | ✓ | ✓ |

---

*Corresponding author  † Equally contributed

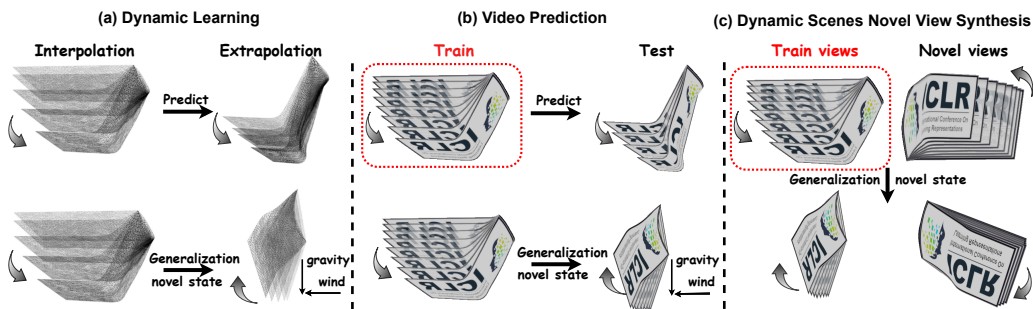

Figure 1: CloDS has the following capabilities:**(a).** Learning underlying cloth dynamics. **(b).** Video prediction through the forward process of DVC. **(c).** Novel view synthesis in dynamic scenes.

spaces of the cloth, the complex physical dynamics, and the strong self-occlusion. Given these challenges, existing visual-only methods struggle to handle CDG effectively. Dynamic scene synthesis methods often fail to generalize beyond observed frames (Hu et al., 2025a; Duan et al., 2024; Zhao et al., 2024; Wang et al., 2025). In contrast, video prediction approaches have predictive capacity, but fall short in CDG due to their difficulty in maintaining temporal consistency amid frequent self-occlusions, and their inability to reason effectively about underlying geometric structures (Gao et al., 2022; Hu et al., 2023). Table 1 presents a comparison of vision-only model capabilities.

Differentiable Visual Computing (DVC) provides a promising solution to CDG by connecting visual observations with their underlying physical representations (Spielberg et al., 2023). As shown in Figure 2, DVC performs video-to-geometry grounding by establishing differentiable mappings between observations and their geometric structures, which enables effective dynamics learning. Existing DVC approaches analogous to CDG include fluid dy-

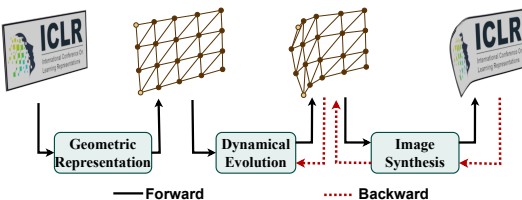

Figure 2: Overview of Differentiable Visual Computing framework in Cloth Dynamics Reasoning.

namics grounding (Guan et al., 2022) and discrete element learner (Wang et al., 2024b), which model fluids and solids with particles. However, particle-based representations are unsuitable for cloth due to its thin structure and small relative position variations. Addressing CDG thus hinges on selecting an appropriate geometric representation and designing an efficient 3D–2D mapping. Given the thin structure and pronounced deformability, we adopt mesh-based representations anchored with Gaussian components (Waczyńska et al., 2024). This configuration effectively establishes differentiable mappings between 3D geometry and 2D observations. Furthermore, complex self-occlusion patterns during cloth motion can induce perspective distortion artifacts in traditional neural rendering (Kerbl et al., 2023). Our framework resolves these challenges through dual-position opacity modulation: simultaneously conditioning Gaussian opacity on world-space (relative) coordinates and mesh-space (absolute) coordinates of Gaussian components.

Therefore, in this paper, we introduce **Clo**th **D**ynamic **S**platting (**CloDS**), the first known unsupervised visual-only method for learning cloth dynamics under unknown conditions. To achieve this goal, we represent the cloth as meshes and design a corresponding Spatial Mapping Gaussian Splatting module with dual-position opacity modulation to establish a differentiable mapping between 2D observations and 3D geometry. Moreover, we develop a three-stage training framework for CloDS to enable unsupervised dynamics learning. The learned dynamics can be applied to downstream animations of various garments (SHAO et al., 2024). Our contributions are summarized as follows:

• We introduce and explore Cloth Dynamics Grounding (CDG), a novel intuitive physical problem.

• We propose Cloth Dynamic Splatting (CloDS) for unsupervised CDG from multi-view videos. CloDS also supports predict video and generate novel view models of dynamic scenes (Figure 1).

• The videos synthesized by CloDS significantly outperform current state-of-the-art video prediction models, and it can also generalize to unseen configuration.

## 2 RELATED WORK

**Mesh-based cloth simulation.** Cloth simulation is a long-standing research field in graphics and physics Lu et al. (2025); Li et al. (2022); Volino et al. (2009); Tiwari & Bhowmick (2023). Mesh-based methods are widely adopted for their geometric accuracy and efficiency Baraff & Witkin (2023); D. Li et al. (2022); Peng et al. (2023); Tiwari et al. (2023). Recent Data-driven approaches are introduced to further enhance simulation efficiency Wandel et al. (2025); Deng et al. (2024); Santesteban et al. (2021); Shao et al. (2023). However, they rely on physics-based supervision from numerical simulators in known environments (e.g., known material parameters, pre-trained GNNs Rong et al. (2025)), and unsupervised dynamics learning in unknown conditions remains an open challenge. Distinct from existing methods, we propose CloDS, a visual-only approach that learns cloth dynamics in an unsupervised manner under unknown conditions.

**Mesh gaussian splatting.** Gaussian splatting has achieved remarkable results in areas such as 3D reconstruction Kerbl et al. (2023); Guédon & Lepetit (2024); Wu et al. (2024b) and Dynamic scenes Novel View Synthesis. While existing work focuses primarily on rigid bodies or minimally deformed objects, scenes with substantial deformations remain understudied Jiang et al. (2024). Current approaches address this problem by applying mesh-based constraints, anchoring Gaussian components to mesh faces and adjusting them as the mesh deforms Waczyńska et al. (2024); Waczynska et al. (2024) Duan et al. (2024); Gao et al. (2024). In CDG, large deformations and self-occlusion cause rendering artifacts (e.g., perspective) if Gaussians remain statically bound. To address this, CloDS dynamically controls opacity using world-space and mesh-space coordinates.

## 3 PROBLEM FORMULATION

In this paper, we aim to build a model for CDG, targeting unsupervised learning of cloth dynamics from multi-view videos. At time $t$, the mesh is denoted as $M_t = (x_t^W, x_t^M, E)$, where $x_t^W, x_t^M \in \mathbb{R}^{K \times 3}$ are the world-space and mesh-space coordinates of $K$ nodes, and $E$ is their connectivity. The corresponding multi-view images are denoted by $Y_t = \{I_t^i\}_{i \in [1,N]}$, where $I_t^i \in \mathbb{R}^{w \times h \times 3}$.

In CDG, the model aims to infer $p(M_{t+1}|M_t)$ recursively over time. Given only multi-view videos $Y_{1:t+1}$, the model needs to learn $p(M_{t+1}|Mt)$ through the estimation of $p(Y_{t+1}|Y_{1:t})$. If the mapping between pixel space and 3D space $p(M_t|Y_{1:t})$ is accessible, $p(Y_{t+1}|Y_{1:t})$ can be expressed as:

$$p(Y_{t+1}|Y_{1:t}) = p(Y_{t+1}|M_{t+1})p(M_{t+1}|M_t)p(M_t|Y_t, Y_{1:t-1}), \qquad (1)$$

where $p(M_t|Y_t, Y_{1:t-1})$ can be framed as a Bayesian filtering problem (Longhini et al., 2025b). Thus, the joint posterior distribution is obtained as follows:

$$p(M_t|Y_t, Y_{1:t-1}) = \eta p(Y_t|M_t)p(M_t|Y_{1:t-1}), \qquad (2)$$

where $\eta$ is a normalization constant ensuring the posterior distribution integrates to 1. If the transition probability function $p(M_{t-1}|Y_{1:t-1})$ can be learned, then $p(M_t|Y_{1:t-1})$ can be expressed as:

$$p(M_t|Y_{1:t-1}) = \int p(M_t|M_{t-1})p(M_{t-1}|Y_{1:t-1}) \, dM_{t-1}. \qquad (3)$$

Therefore, to iteratively learn $p(Y_{t+1}|Y_{1:t})$, it is essential to model the dynamic transition $p(M_{t+1}|M_t)$ and spatial mappings $p(Y_t|M_t)$, $p(M_t|Y_{1:t})$. These models are jointly trained to approximate $p(Y_{t+1}|Y_{1:t})$ (see Appendix A for details). After learning $p(M_{t+1}|M_t)$, the model can simulate cloth dynamics. Simultaneously, using $p(Y_t|M_t)$, it maps the 3D underlying representation of the cloth to the pixel space, forming a continuous video to implement DVC forward process.

## 4 METHOD

In this section, we describe the Cloth Dynamics Splatting framework. We first delineate the overall methodology, then detail the neural simulator and the Spatial Mapping Gaussian Splatting module, and finally describe the training procedure.

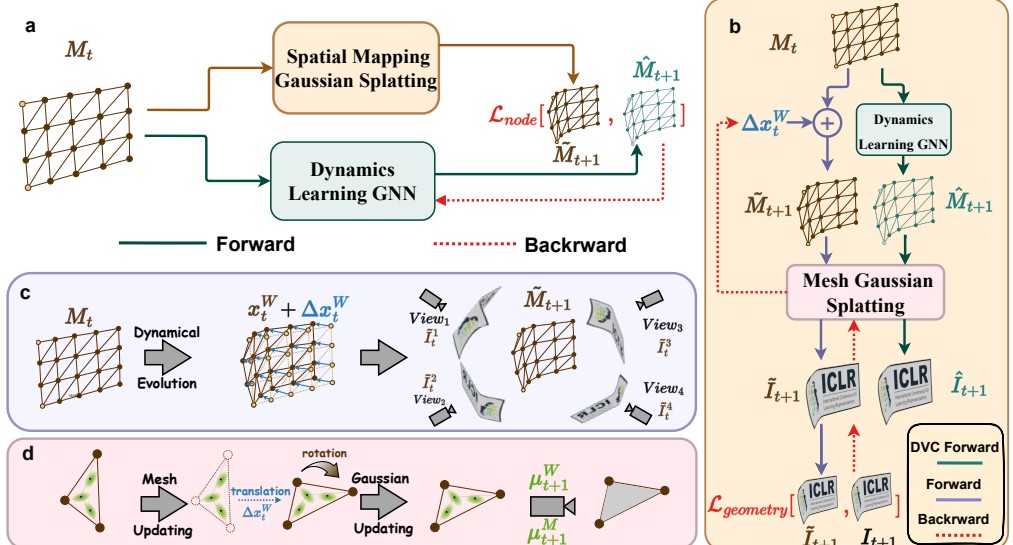

Figure 3: The overview of Cloth Dynamics Splatting (CloDS) for Cloth Dynamics Grounding. **(a).** Overall model architecture. **(b).** The forward and backward processes of Spatial Mapping Gaussian Splatting (SMGS) and DVC forward process. **(c).** The detail of SMGS forward process. **(d).** The Mesh Gaussian Splatting in SMGS. SMGS can obtain the mesh of cloth $\tilde{M}_{t+1}$ based on $M_t$ or $\tilde{M}_t$ through backpropagation. In the forward stage of DVC, $\Delta x_t^W$ is typically set to $\mathbf{0}$. However, when extracting the mesh $\tilde{M}_{t+1}$ from images $I_{t+1}$, the learnable $\Delta x_t^W$ is used to make $\tilde{I}_{t+1}$ close to $I_{t+1}$.

## 4.1 OVERVIEW

Most existing methods rely on physics-based supervision in known environments, while unsupervised dynamics learning under unkonwn condition remains a key challenge. Unlike existing methods, we directly learn cloth dynamics from visual observations. To achieve this goal, we design Spatial Mapping Gaussian Splatting (Section 4.3), a differentiable 2D-to-3D mapping module that grounds video frames into geometric representations. This mapping permits the subsequent Dynamics Learning GNN (Section 4.2) to learn cloth dynamics directly from pixel-space supervision. An overview of our approach is depicted in Figure 3.

## 4.2 GRAPH NEURAL DYNAMICS LEARNER

Due to the thin-plane and self-occlusion properties of cloth, we model it as meshes. The dynamic learner $p(M_{t+1}|M_t)$ recursively estimates mesh $\hat{M}_{1:T}$ from an initial state $M_0$. Conventional mesh-based prediction methods typically employ Graph Neural Networks (GNNs) to model dynamics. Our framework supports various GNN architectures. Without loss of generality, we select MGN (Pfaff et al., 2020) as neural simulator. MGN encodes **world-space coordinates** $x_t^W$, **mesh-space coordinates** $x_t^M$, and relative positions, then applies message passing to model interactions among nodes. $x_t^W$ denotes the 3D spatial coordinates, and $x_t^M$ denotes the 2D UV coordinates. The aggregated node features are decoded to the next positions $x_{t+1}^W$. As mesh targets are not available in CDG, an effective mapping function is required to extract meshes from the pixel space.

## 4.3 SPATIAL MAPPING GAUSSIAN SPLATTING

Gaussian Splatting (GS) represents the radiance field with anisotropic 3D Gaussians. Each Gaussian component is characterized by a center $\mu$ and a covariance matrix $\sum$ which is decomposed into a rotation $R$ and a scale $S$. The color of each component is determined by a spherical harmonic function $c_i$. Pixel color $C$ is obtained by compositing $N$ components as $\sum_{i \in N} c_i \alpha_i \prod_{j=1}^{i-1}(1 - \alpha_j)$, where $\alpha_i$ is the opacity of components. Gaussian Splatting enables 3D-to-pixel mapping. However, CDG requires temporally consistent mappings to learn dynamics from the pixel space. To address this chal-

lenge, we develop Spatial Mapping Gaussian Splatting (SMGS) based on the GaMes (Waczyńska et al., 2024) which leverage meshes to controllably preserve the Gaussian correspondence.

We first establish a static mapping by anchoring Gaussian components on the mesh faces. Each Gaussian center $\mu_t$ is computed via barycentric interpolation: $\mu_t = \beta_1 X_{t,1}^W + \beta_2 X_{t,2}^W + \beta_3 X_{t,3}^W$, where $\beta_1 + \beta_2 + \beta_3 = 1$ and $V = [X_{t,1}^W, X_{t,2}^W, X_{t,3}^W]$ is the face. As the mesh deforms, Gaussians are updated by the same $\beta$ with new vertex positions $X_t^W + \Delta X_t^W$ to maintain temporal consistency mapping. The rotation matrix of each Gaussian, $\mathbf{R} = [\mathbf{r}_1, \mathbf{r}_2, \mathbf{r}_3]$, must also remain consistent with the rotation of its associated face. The first vertex of $\mathbf{R}$ is defined by normal vector:$\mathbf{r}_1 = \text{norm}\big((X_{t,2}^W - X_{t,1}^W) \times (X_{t,3}^W - X_{t,1}^W)\big)$, where $\times$ is the cross product. The second vertex is $\mathbf{r}_2 = \text{norm}\big((X_{t,2}^W - X_{t,1}^W)\big)$. The third is obtained as a single step in the Gram–Schmidt process (Strang, 2000): $\mathbf{r}_3 = \text{norm}\big(\text{orth}\big(X_{t,3}^W - X_{t,1}^W, \mathbf{r}_1, \mathbf{r}_2\big)\big)$.

The scale $S = \text{diag}(s_1, s_2, s_3)$, where $s_1 = \epsilon$, $s_2 = \|X_{t,2}^W - X_{t,1}^W\|$ and $s_3 = \langle X_{t,3}^W - X_{t,1}^W, r_3 \rangle$. However, in CDG, strong self-occlusions and large deformations cause existing mesh-based GS methods suffering from perspective distortions and color errors (Fig-

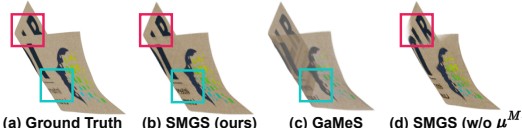

(a) Ground Truth  (b) SMGS (ours)  (c) GaMeS  (d) SMGS (w/o $\mu^M$)

Figure 4: The result of rendering. Videos are available in Part 2 at URL.

ure 4c). To address this, we modulate the Gaussian opacity through dual-position opacity modulation based on the world-space $\mu^W$ and mesh-space coordinates $\mu^M$ of their centers $\mu$ (Figure 3d). The dual-position opacity modulation is defined as:

$$\alpha_{i,t} = f_\theta(\mu_{i,t}^W, \mu_{i,t}^M), \tag{4}$$

where $f_\theta$ is a Multilayer Perceptron with parameters $\theta$, and $\mu_{i,t}^W$, $\mu_{i,t}^M$ represent the world-space (relative positions) and mesh-space coordinates (absolute positions) of $i_{th}$ Gaussian center. Detailed explanations of absolute and relative coordinates are provided in Appendix B. Without world-space coordinates $\mu^W$, the dual-position opacity modulation reduces to standard GaMeS. Relative positions reduce perspective errors (Figure 4c) and absolute positions $\mu^M$ prevent the cloth from becoming transparent when it moves into previously unseen regions (Figure 4d). This mapping enables 3D-to-2D projection in CDG, allowing CloDS to complete the DVC forward rendering (Figure 3c). After establishing temporal Gaussian correspondence, we recover 3D labels for the neural simulator by mapping back from 2D. Inspired by 3D reconstruction (Worchel et al., 2022), we backpropagate $\Delta x_t^W$ to adjust mesh nodes from $x_t^W$ to $\tilde{x}_{t+1}^W$, aligning the rendered image with $Y_{t+1}$ (Figure 3b):

$$\arg \min_{\Delta x_t^W} \mathcal{L}_{geometry}(\text{SMGS}(\tilde{x}_{t+1}^W), \tilde{x}_{t+1}^W, x_0^W, Y_{t+1}), \tag{5}$$

where $\mathcal{L}_{geometry}$ is the reconstruction loss (detailed in Section 4.4), and $\tilde{x}_{t+1}^W = \tilde{x}_t^W + \Delta x_t^W$. Gaussian components are then updated and re-rendered until convergence, establishing the 2D-to-3D mapping and enabling recursive 3D label generation for training the Dynamics Learning GNN.

## 4.4 DYNAMICS-AWARE UNSUPERVISED TRAINING FRAMEWORK

To solve CDG, we propose a three-step training framework. First, we estimate SMGS $p(Y_t|M_t)$. Then, SMGS maps images to 3D space via backpropagation to obtain $p(M_t|Y_{1:t})$, recursively reconstructing $\tilde{M}_{1:T}$. Finally, the dynamics simulator is trained with $\tilde{M}_{1:T}$ to model $p(M_{t+1}|M_t)$. The first two stages adopt single-step training without temporal losses, while the third stage applies a rollout strategy for temporal learning. The overall algorithm is summarized in Algorithm S.1.

**First stage: Gaussian component construction.** The mesh and multi-view images of the first frame are used to build the cloth's Gaussian component representation via SMGS, optimized with the standard 3D Gaussian splatting loss (Kerbl et al., 2023):

$$\mathcal{L}_{render} = (1 - \lambda)\mathcal{L}_1(\tilde{Y}_t, Y_t) + \lambda\mathcal{L}_{D-ssim}(\tilde{Y}_t, Y_t), \tag{6}$$

where $\lambda$ is a constant (typically 0.2), $\tilde{Y}_t$ is the SMGS rendering result at time $t$, and $Y_t$ is the ground truth. Equation 6 is also used as part of the second stage loss, so we use $\tilde{Y}$ instead of $\tilde{Y}_0$. Unlike the warm-up phase in NeuroFluid (Guan et al., 2022), which requires particle representations across

multiple time steps to train the rendering model, CloDS only needs the mesh from the first frame to construct the 3D cloth representation.

**Second stage: Extracting mesh from image space.** By iteratively optimizing $\Delta x_t^W$ in SMGS $P(Y_{t+1}|M_t)$, we recursively obtain the mesh $\tilde{M}_{1:T}$. An edge loss is introduced during training to preserve cloth shape and prevent excessive deformation by maintaining relative node distances:

$$\mathcal{L}_{geometry} = \mathcal{L}_1(\text{SMGS}(\tilde{x}_{t+1}^W), Y_{t+1}) + \gamma \mathcal{L}_{edge}, \tag{7}$$

$$\mathcal{L}_{edge} = \sum_{i=0}^{N-1} \sum_{[i,j]\in E} \left| d(\tilde{x}_{i,t+1}^W, \tilde{x}_{j,t+1}^W) - d(x_{i,0}^W, x_{j,0}^W) \right|, \tag{8}$$

where $\gamma$ is a constant, $x_0^W$ is the world-space coordinate of the nodes in the first frame, $N$ is the number of nodes and $\tilde{x}_{t+1}^W = \tilde{x}_t^W + \Delta x_t^W$ is the node coordinates obtained through SMGS mapping.

**Third stage: dynamics simulator training.** Finally, the meshes $\tilde{M}_{1:T}$ obtained from the 2D pixel-space serve as supervision for training the GNN, allowing it to learn cloth dynamics under unknown conditions. We adopt the loss function from (Pfaff et al., 2020) with a rollout strategy:

$$\mathcal{L}_{node} = \sum_{t=1}^{T} \text{MSE}(\hat{x}_t^W, x_t^W), \tag{9}$$

where $T$ is the rollout length (set to 8 in our experiments) and $\hat{x}_t^W = \text{GNN}(...(\text{GNN}(x_0^W))...)$. After training, GNN learns the cloth dynamics. Combined with SMGS, it enables dynamical evolution and image synthesis (Algorithm S.2), completing the DVC forward process shown in Figure 1.

## 5 EXPERIMENT

In this section, we evaluate the performance of CloDS and its components: Dynamics Learning GNN and SMGS. We first describe the experimental setup. We then assess CloDS's CDG performance, demonstrate SMGS effectiveness in dynamic scene novel view synthesis, and evaluate CloDS in the DVC forward process. Additionally, we explore CloDS's generalization and real-world potential, and finally, we further analyze the impact of neural simulator selection and SMGS components.

### 5.1 EXPERIMENTAL SETUP

**Dataset and metrics.** To evaluate whether the model has learned the underlying dynamics, we follow existing video-based dynamics learning works (Wang et al., 2024b; Zhu et al., 2024a) to generate the dataset. We generate the benchmark using Blender (Community, 2018) on the FLAGSIMPLE dataset (Pfaff et al., 2020) by rendering multi-view cloth videos. FLAGSIMPLE training dataset consists 1000 trajectories (400 steps each), with unique initial states. We render multi-view images for 120 trajectories: 100 for training (**viewed**) and 20 for testing (**unviewed**). Detailed dataset information is provided in Appendix D.1. CloDS is evaluated on Cloth Dynamics Grounding, Dynamic Scene Novel View Synthesis, and DVC forward modeling, with splits detailed in Appendix D.1.

For consistent comparisons, we use task-specific metrics. Following NeuroFluid (Guan et al., 2022), CDG uses rollout RMSE between the predicted and ground truth node positions. Dynamic Scene Novel View Synthesis adopts PSNR, SSIM (Wang et al., 2004), and LPIPS (Zhang et al., 2018) following 3DGS (Kerbl et al., 2023). The DVC forward process reports PSNR, SSIM, LPIPS, and RMSE following video prediction works (Zhong et al., 2023). Metric details are in Appendix D.2.

**Baseline models.** In CDG, a fully fair comparison with mesh-based training methods is challenging, as CloDS targets unsupervised cloth dynamics learning. We adopt MGN (Pfaff et al., 2020) as the Dynamics Learning GNN and use it as a baseline, comparing CloDS trained on video data with MGN trained on mesh data to evaluate whether CloDS can effectively learn cloth dynamics from visual observations. For Dynamic Scene Novel View Synthesis, we compare SMGS with GaMeS (Waczyńska et al., 2024), as well as dynamic scene novel view synthesis models: 4DGS (Wu et al., 2024a), MSTH (Wang et al., 2024a) and M5D-GS (Hu et al., 2025b). For the DVC forward process, we compare CloDS with video prediction models: SimVP (Gao et al., 2022), MAU (Chang et al., 2021), MMVP (Zhong et al., 2023), and TAU (Tan et al., 2023). See Appendix D.3 for details.

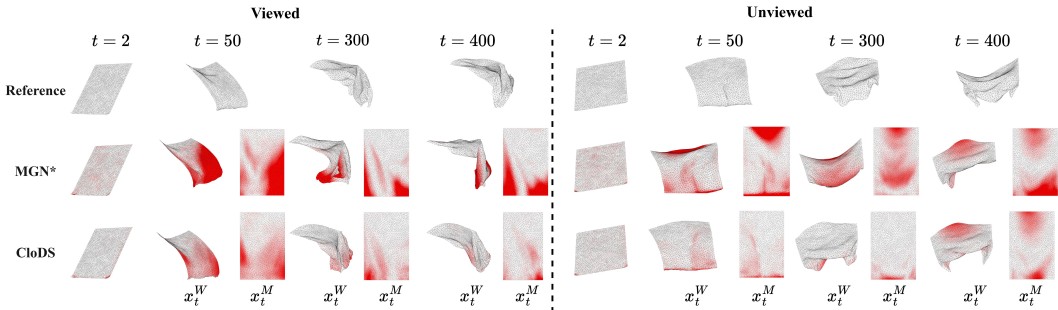

Figure 5: Visualization of the cloth prediction results. $t = 2$ denotes the first predicted frame, $t = 300$ is the last frame of interpolation, and $t = 400$ is the last frame of extrapolation. $x_t^W$ and $x_t^M$ represent the 3D and 2D mesh positions, respectively. Errors are visualized in red, with deeper color indicating larger errors. All methods share the same error bars. "Viewed" and "Unviewed" denote whether the cloth's initial state was seen during training.

Table 2: Average RMSE between predicted mesh nodes and ground truth (details in Appendix D.2). "Viewed" and "Unviewed" refer to trajectories 1–50 in the training set and all trajectories in the test set, respectively. Mean and standard deviation are reported.

| Method | Viewed | | | | | | Unviewed | | | | | |
|---|---|---|---|---|---|---|---|---|---|---|---|---|
| | Interpolation | | | Extrapolation | | | Interpolation | | | Extrapolation | | |
| | $d_{t\leq300}^{\text{AVG}}$ | $d_{t=300}$ | Avg. | $d_{300<t\leq400}^{\text{AVG}}$ | $d_{t=400}$ | Avg. | $d_{t\leq300}^{\text{AVG}}$ | $d_{t=300}$ | Avg. | $d_{300<t\leq400}^{\text{AVG}}$ | $d_{t=400}$ | Avg. |
| MGN | 0.1293 | 0.1279 | 0.1286 | 0.1304 | 0.1277 | 0.1291 | 0.1357 | 0.1359 | 0.1358 | 0.1329 | 0.1299 | 0.1314 |
| | ± 0.024 | ± 0.031 | ± 0.028 | ± 0.026 | ± 0.033 | ± 0.027 | ± 0.035 | ± 0.029 | ± 0.034 | ± 0.032 | ± 0.026 | |
| MGN* | 0.1394 | 0.1366 | 0.1380 | 0.1386 | 0.1390 | 0.1388 | 0.1426 | 0.1493 | 0.1460 | 0.1394 | 0.1329 | 0.1362 |
| | ± 0.084 | ± 0.078 | ± 0.071 | ± 0.069 | ± 0.073 | ± 0.067 | ± 0.095 | ± 0.059 | ± 0.072 | ± 0.088 | ± 0.064 | |
| CloDS* | 0.1313 | 0.1305 | 0.1309 | **0.1312** | 0.1336 | 0.1324 | 0.1437 | 0.1418 | 0.1428 | 0.1381 | 0.1411 | 0.1396 |
| | ± 0.046 | ± 0.039 | ± 0.044 | ± 0.037 | ± 0.048 | ± 0.033 | ± 0.029 | ± 0.045 | ± 0.037 | ± 0.046 | ± 0.048 | |
| CloDS** | **0.1302** | **0.1285** | **0.1294** | 0.1320 | **0.1293** | **0.1307** | **0.1395** | **0.1381** | **0.1388** | **0.1340** | **0.1310** | **0.1325** |
| | ± 0.025 | ± 0.038 | ± 0.023 | ± 0.036 | ± 0.034 | ± 0.037 | ± 0.048 | ± 0.044 | ± 0.037 | ± 0.039 | ± 0.045 | |
| CloDS | 0.1320 | 0.1322 | 0.1321 | 0.1349 | 0.1338 | 0.1344 | 0.1408 | 0.1390 | 0.1399 | 0.1353 | 0.1324 | 0.1339 |
| | ± 0.065 | ± 0.057 | ± 0.073 | ± 0.058 | ± 0.066 | ± 0.059 | ± 0.064 | ± 0.058 | ± 0.046 | ± 0.079 | ± 0.083 | |

## 5.2 CLODS ENABLES UNSUPERVISED CLOTH DYNAMICS LEARNING FROM VIDEOS

We train models on different subsets of mesh and video data. We use all training mesh trajectories and the first 50 trajectories to obtain **MGN** and **MGN\***, respectively. We then fine-tuned MGN* on the remaining 50 training videos to obtain **CloDS**. Finally, we trained **CloDS\*** on the first 50 videos and **CloDS\*\*** on all training videos. We conduct interpolation and extrapolation experiments on both training and test sets. In NeuroFluid (Guan et al., 2022), interpolation of viewed states is termed dynamic grounding, extrapolation of viewed states is termed dynamic prediction and prediction of unviewed states is termed novel scene generalization. The results are shown in Table 2.

First, CloDS consistently outperforms MGN* on both viewed and unviewed trajectories. As final frame predictions accumulate rollout errors, this highlights the superiority of CloDS in capturing cloth dynamics. Figure 5 visualizes the final-frame predictions. CloDS aligns more closely with the ground truth, particularly in challenging regions such as cloth edges, demonstrating that our approach effectively enables unsupervised dynamics learning. Secondly, CloDS* achieves better

Table 3: Performance on Dynamic Scenes Novel View Synthesis.

| Model | PSNR (dB)↑ | SSIM$_{\times10}$↑ | LPIPS$_{\times1000}$↓ |
|---|---|---|---|
| 3DGS | 39.6263±3.17 | 9.986±0.07 | 2.53±0.20 |
| 4DGS | 23.2089±1.86 | 9.718±0.06 | 15.82±1.27 |
| MSTH | 23.1353±1.85 | 9.682±0.08 | 16.53±1.32 |
| M5D-GS | 29.3428±2.35 | 9.731±0.13 | 12.97±1.04 |
| GaMeS | 33.0249±1.54 | 9.937±0.12 | 5.21±0.42 |
| SMGS (Ours) | **36.2368**±1.17 | **9.959**±0.06 | **3.53**±0.21 |

performance on viewed trajectories than CloDS but worse on unviewed ones. It is attributed to overfitting on seen trajectories and limited ability to generalize to unseen trajectories because of the insufficient training data. However, it confirms that our approach can effectively learn dynamics from videos in an unsupervised manner. Finally, CloDS** achieves performance close to MGN, which relies on full mesh supervision. This demonstrates that with sufficient data, CloDS enables near-optimal cloth dynamics learning (*i.e.*, MGN trained on all mesh trajectories). We further analyze the impact of initial frame errors on CloDS's ability to learn cloth dynamics in Appendix E.

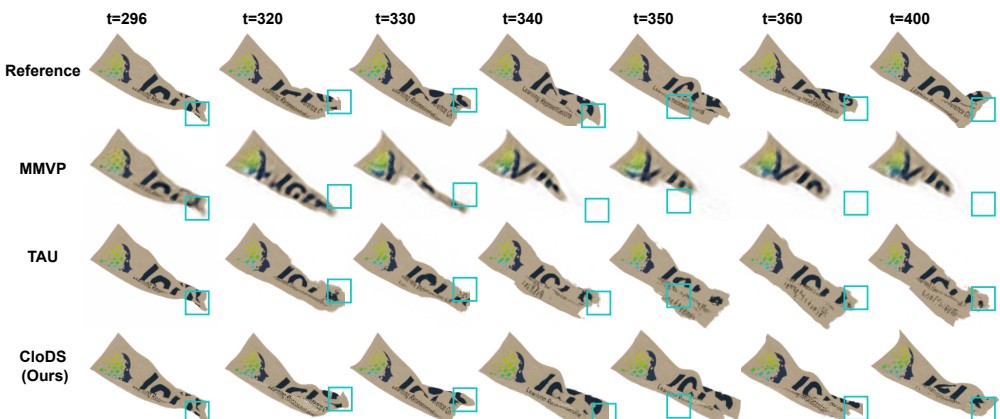

Figure 6: Visualization of DVC process and video prediction results. Videos are in Part 1 at URL.

### 5.3 SMGS IMPROVES RENDERING UNDER DEFORMATION AND OCCLUSION

The results of dynamic scene novel view synthesis are shown in Table 3. 3DGS, trained separately per frame, serves as the upper bound. It is evident that SMGS achieves superior performance in novel view synthesis. This is primarily due to the large deformation and strong self-occlusion caused by cloth motion, which existing models fail to handle adequately. GaMeS does not adjust the opacity of the Gaussian components during cloth motion, leading to incorrect weight distribution when components overlap in occluded cloth areas. This causes perspective and rendering errors (Figure 4c). In contrast, SMGS assigns opacity using both 3D world-space and mesh-space coordinates: the former preserves relative positioning to prevent perspective errors, while the latter constrains opacity to avoid transparency in unseen regions (Further ablation study on SMGS detailed in Section 5.8).

### 5.4 CLODS OUTPERFORMS ON THE FORWARD PROCES OF DVC

Video prediction models learn physical dynamics from videos by generating future frames from current observations without reasoning about 3D structures. CloDS also trains on videos. As shown in Figure 3b, it predicts 3D mesh states, from which we render cloth-

Table 4: Performance on cloth dynamic video prediction.

| Model | PSNR (dB)↑ | $SSIM_{\times 10}$↑ | $LPIPS_{\times 100}$↓ | $RMSE_{\times 100}$↓ |
|---|---|---|---|---|
| MAU | 23.7249±1.82 | 9.745±0.06 | 1.548±0.11 | 6.78±0.54 |
| TAU | 23.9968±1.88 | 9.781±0.06 | 1.438±0.10 | 6.65±0.51 |
| MMVP | 23.9678±1.86 | 9.770±0.06 | 1.335±0.10 | 6.59±0.50 |
| SimVP | 25.4770±2.04 | 9.801±0.07 | 1.020±0.09 | 5.57±0.45 |
| CloDS (ours) | **26.6207**±0.39 | **9.817**±0.03 | **0.899**±0.04 | **4.78**±0.21 |

motion videos using SMGS. Table 4 compares CloDS with existing video prediction models, where CloDS achieves significantly higher video quality. To investigate this result, we present visualized outputs in Figure 6 (video demonstrations are available in Part 1 at URL). These visualizations reveal that video prediction models accumulate errors over time at cloth edges due to strong self-occlusion, which hampers their ability to maintain temporal consistency. In contrast, CloDS directly models the cloth in 3D space, thereby ensuring better consistency in self-occlusion regions. We further analyze the performance of CloDS and video prediction models on the multi-trajectories in Appendix H.1.

### 5.5 FUTURE ANALYSIS

**CloDS Generalizes to New Shapes and Textures.** The ability of a neural simulator to generalize to unseen cloth shapes reflects its effectiveness in learning cloth dynamics. We evaluate this by testing our video-supervised dynamic simulator on a cylindrical cloth (Figure 7a). The results show accurate predictions on the new shape, indicating that CloDS successfully learns cloth dynamics from videos in an unsupervised manner. To assess texture robustness, we retrain CloDS with modified cloth textures. As shown in Figure 7b, CloDS maintains strong performance despite texture changes. Videos are available in Part 3 and 4 at URL. Additional results on learning cloth dynamics under complex lighting conditions are presented in Appendix H.5.

**CloDS has the potential for application to real-world datasets.** This paper investigates the feasibility of addressing CDG. Synthetic data facilitates training and evaluation with ground truth meshes. Most existing methods for unsupervised dynamics learning from videos rely on simulated rather

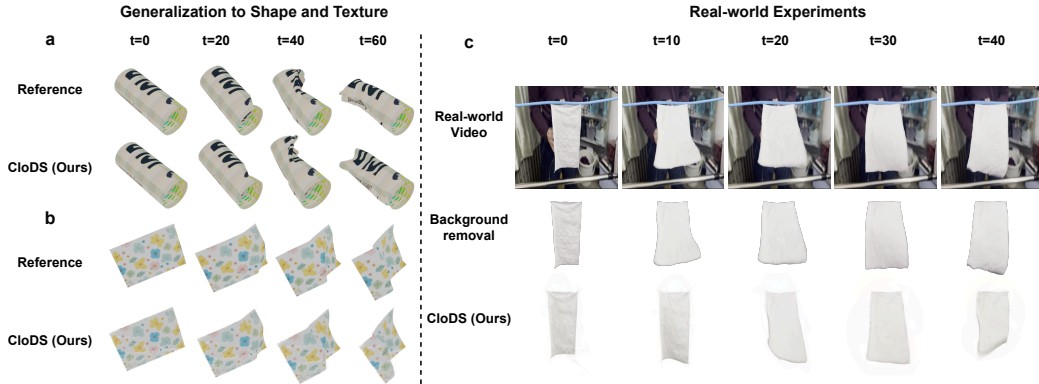

Figure 7: Generalization capabilities of CloDS. **(a).** Cloth shape generalization. **(b).** Cloth texture robustness. **(c).** Real-World data exploration. Videos are available in Part 3,4 and 5 at URL.

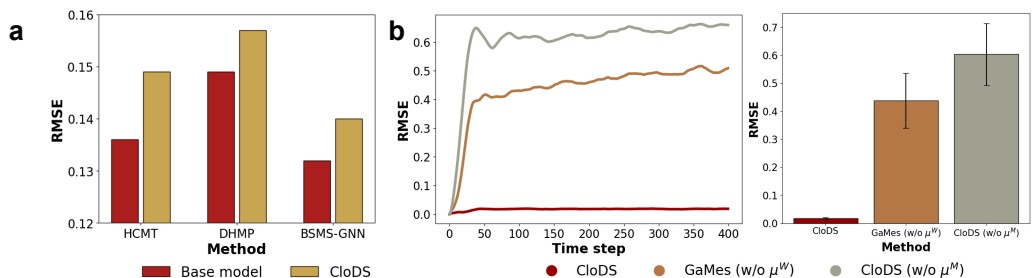

Figure 8: Ablation study. **(a).** Impact of neural dynamic simulator replacement. **(b).** Impact of dual-position opacity modulation in SMGS.

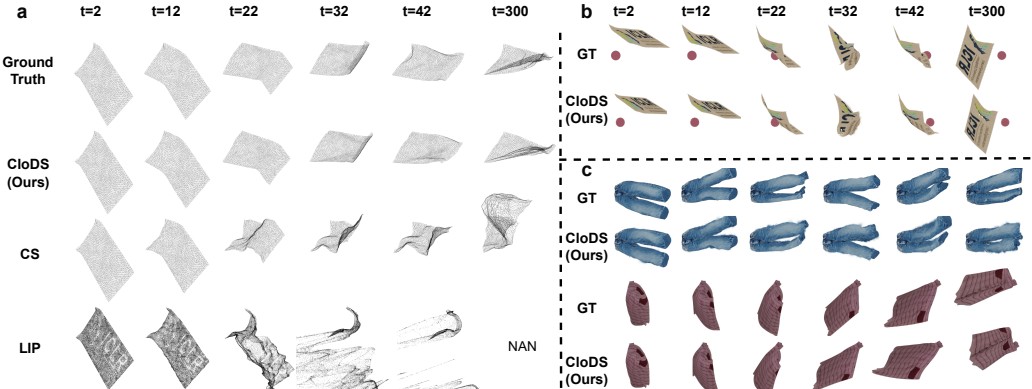

Figure 9: Visualization of result. **(a).** Cloth prediction results. "NAN" is no valid output. **(b).** Object-cloth collision. **(c).** Real-world garment. Videos are available in Part 8, 9 and 10 at URL.

than real data (Wang et al., 2024b; Guan et al., 2022). However, we acknowledge that real-world exploration is meaningful and challenging (*e.g.,* ensuring experiment consistency and dataset construction). To this end, we make our best efforts for CloDS in real-world settings. We first capture multi-view videos of cloth. To focus CloDS on learning cloth dynamics, we extract the cloth regions by SAM (Kirillov et al., 2023) and then train CloDS with the extracted videos. Demo videos are available in Part 5 at URL. We observe that CloDS is able to ground cloth dynamics from real-world data, although artifacts remain. This may be attributed to camera frame rate limitations and complex real-world lighting conditions. We leave further improvements to future work.

## 5.6 CLoDS PROVIDES ADVANTAGES OVER EXISTING GEOMETRY-AWARE APPROACHES

Since CDG fundamentally relies on 3D-aware modeling, it is essential to compare against geometry-aware and unsupervised methods. we adapt LIP Zhu et al. (2024b), which uses point clouds as geometric representation, to CDG setting. We also adapt CS Longhini et al. (2025a), which represents cloth using meshes. It is important to note that neither LIP nor CS is fully unsupervised. For LIP, we pretrain Particle Posterior Estimator and the Probabilistic Particle Simulator on cloth point-cloud data. For CS, we pretrain GNN on cloth-mesh data. Both LIP and CS are then trained on videos. The predicted 3D geometry is visualized in Figure 9a. We observe that our method substantially outperforms existing geometry-aware approaches. LIP employs point clouds as its geometric representation, and the lack of mesh-based topological constraints makes it difficult to preserve stable relative positions during inference, leading to a rapid collapse of the cloth shape. In contrast, CS benefits from the mesh topology, which enforces more coherent and physically plausible deformations. As a result, it performs noticeably better than LIP, though still falling short of CloDS.

## 5.7 CLoDS EXCELS UNDER COMPLEX FORCES AND REAL-WORLD GARMENT

**CloDS Demonstrates Robust in Object-Cloth Collision Dynamics.** CloDS is designed to learn cloth dynamics in an unsupervised manner under unknown environments. In this section, we investigate its applicability to more complex scenarios containing multiple interacting objects. Inspired by VGPL Li et al. (2020), we assign each object a rigidity attribute, which constrains the allowable relative motion among its nodes. This additional attribute enables the model to handle diverse materials and their interactions. To systematically train and evaluate, we construct an Object–Cloth Collision dataset, where a piece of cloth falls under gravity and interacts with a moving rigid sphere. We then train CloDS on this dataset. As shown in Figure 9b, CloDS successfully learns the underlying object–cloth collision dynamics, demonstrating strong generalization to multi-object scenarios.

**CloDS Generalizes Well to Real Garment Dynamics.** In Section 5.5, we demonstrated the potential of CloDS on real-world datasets. Here, we further evaluate its generalization to real garments. We construct a Real-Garment dataset by performing physics-based simulations Narain et al. (2012) on high-quality garment meshes from the DeepFashion3D V2 dataset Heming et al. (2020). CloDS is then trained and evaluated on this dataset. As shown in Figure 9c, CloDS learns reliable dynamics even on realistic garment geometries, indicating strong generalization to complex real-world shapes.

## 5.8 ABLATION STUDY

**Impact of neural simulator selection.** The neural simulator is designed to learn cloth dynamics. Without loss of generality, we adopt MGN as the backbone. To further demonstrate the robustness of CloDS, we also use HCMT (Yu et al., 2024), DHMP (Deng et al., 2024), and BSMS-GNN (Cao et al., 2023) as backbones. As shown in Figure 8a, all based models trained only on nodes show performance improvement after further training with videos, confirming the robustness of CloDS.

**Impact of SMGS components.** To validate the necessity of introducing $\mu^W$ and $\mu^M$ in SMGS, we conduct ablation study during the mesh extraction stage (Second Stage). As shown in Figure 8b, errors accumulate over time without $\mu^W$ and $\mu^M$, eventually making the meshes unusable. This is because, without $\mu^W$ and $\mu^M$, perspective errors and transparency issues in unseen regions occur, respectively. Visualizations of the errors are available in Part 2 at URL.

## 6 CONCLUSION

This paper explores a novel problem: Cloth Dynamics Grounding, where cloth dynamics are learned from visual observations under unknown conditions without physical supervision. To address this challenge, we propose CloDS together with three-stage training framework. CloDS utilizes Spatial Mapping Gaussian Splatting to establish a mapping between the 2D pixel space and 3D space, enabling the dynamic simulator to learn cloth dynamics directly from videos. SMGS handles large deformations and severe self-occlusion by using both relative and absolute positions of the Gaussian components. This design ensures an accurate mapping between the 2D and 3D spaces during rendering. Extensive experiments confirm the necessity of this design. They also show that CloDS effectively addresses CDG and outperforms existing models in dynamic novel view synthesis and video prediction tasks. In future work, we aim to explore visual learning of the dynamics of multiple objects in complex scenes under unknown conditions.

ETHICS STATEMENT

We confirm that all authors have read and adhered to the ICLR Code of Ethics. This work complies with the ethical principles outlined therein, and no part of this study violates ethical standards.

REPRODUCIBILITY STATEMENT

We have made every effort to ensure the reproducibility of our results. A complete description of our model architecture, training procedure and evaluation metrics is our paper, with additional implementation details in Appendix D. Our source code, including data preprocessing scripts and configuration files, is available at https://github.com/whynot-zyl/CloDS. This repository enables the reproduction of all experiments reported in the paper.

THE USE OF LARGE LANGUAGE MODELS

In preparing this manuscript, we used a Large Language Model (ChatGPT, GPT-5) solely for language refinement and grammar editing. All research ideas, experimental designs, analyses, and interpretations were conceived and executed by the authors. The model did not contribute to research ideation, data collection, analysis, or the generation of novel scientific content. All scientific claims, interpretations, and conclusions are entirely our own, and the authors take full responsibility for the contents of this paper.

ACKNOWLEDGMENTS

The work is supported by the National Natural Science Foundation of China (No. 62276269 and No. 92270118), the Beijing Natural Science Foundation (No. 1232009), and the Strategic Priority Research Program of the Chinese Academy of Sciences (No. XDB0620103).

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

# APPENDIX

## A   PROBLEM FORMULATION DETAILS

Based on Eq. 2 and Eq. 3, the problem can be defined as solving the following equation:

$$p(Y_{t+1}|Y_{1:t}) = \eta p(Y_{t+1}|M_{t+1})p(M_{t+1}|M_t)p(Y_t|M_t) \cdot \int p(M_t|M_{t-1})p(M_{t-1}|Y_{1:t-1}) \, dM_{t-1}. \tag{S.1}$$

We can observe that by solving this equation, the model can learn the joint posterior distribution $p(M_{t+1}|M_t)$.

From Eq. S.1, we can see that in order to learn $p(Y_{t+1}|Y_{1:t})$ by iterative method, the key is to model the dynamics learning function $p(M_{t+1}|M_t)$ and the spatial mapping function $p(Y_t|M_t)$, $p(M_t|Y_{1:t})$. The dynamics learning function $p(M_{t+1}|M_t)$ primarily serves to learn cloth dynamics, predicting the next mesh state $\hat{M}_{t+1}$ based on the current mesh state $M_t$. The spatial mapping function $p(Y_t|M_t)$ and $p(M_t|Y_{1:t})$ establish a differentiable mapping between 3D real-world space and 2D pixel space, bridging the gap between the underlying representation of cloth and real-world observations.

## B   DETAILED EXPLANATIONS OF ABSOLUTE AND RELATIVE COORDINATES

In this paper, the distinction between absolute and relative is determined by whether a node's position varies in space over time. The position of node **changes** in world space, so world-space coordinate $x^W$ is **relative**. The position of node does **not change** in mesh space, so mesh-space coordinate $x^M$ is **absolute**.

We build the Gaussian representation of cloth from the first frame. During dynamics learning, the cloth then moves to new poses. If we use only $X^M$ and not $X^W$ in Equation 4, the opacity of the Gaussian components does not change with the relative position during the deformation of cloth. This reduces s to standard GaMeS (Waczyńska et al., 2024). Fixed opacity causes perspective errors during rendering (Figure 4c). Adding $X^W$ allows the opacity to change according to the relative position of the components and avoids perspective problems. If we use only $X^W$ and not $X^M$ in Equation 4, the Gaussian components obtain incorrect opacity in regions not covered in the first frame. This results in components not being visible in the observations (Figure 4d). Adding $X^M$ as a fixed variable maintains robustness to new 3D positions during rendering.

## C   CLOTH DYNAMICS SPLATTING ALGORITHM

We train CloDS using a three-stage training framework (Section 4.4), as detailed in Algorithm S.1. Upon completion of training, the learned Gaussian representation and the neural simulator GNN are employed to predict mesh dynamics, thereby realizing the forward process of DVC. The full generation pipeline for this forward process is illustrated in Algorithm S.2.

## D   EXPERIMENT DETAILS

### D.1   DATASET DETAILS

The FLAGSIMPLE dataset was simulated using ArcSim (Narain et al., 2012) with linear elements. Its training set contains 1,000 trajectories, 100 validation, and 100 test trajectories. The initial state of each trajectory is unique, and the cloth meshes are regular which means all edges having similar length. Each trajectory represents the dynamic evolution of the same shaped cloth in the same environment over 400 time steps. The simulation time step is 0.02s. The cloth meshes are triangular and regular which means all edges have similar lengths. We used Blender to render all time steps of the first 120 trajectories from the FLAGSIMPLE dataset training set to generate the training and testing images with a resolution of $800 \times 800$ in this paper. During Blender rendering, we used the first frame of the first trajectory to extract the cloth's UV coordinates, and applied the same UV coordinates to the mesh for all time steps, ensuring consistency in the texture of all cloths.

---

**Algorithm S.1** Dynamics-Aware Unsupervised Training Framework.

---

**Input:** Sequences of image from $N$ different viewpoints $Y_{1:T+1} = \{I^i_{1:T+1}\}_{i \in [1,N]}$. Mesh is denoted as $M = (x^W, x^M, E)$.

1: Extracting initial Mesh $M_1$ from the initial frame $I^{1:N}_1$ via 2D Gaussian Splatting Huang et al. (2024).
2: **repeat**                                                  ▷ Gaussian Component Construction
3:     $\tilde{I}^{1:N}_1 \leftarrow SMGS(M_1, View_{[1:N]})$.
4:     Compute render loss $\mathcal{L}_{geometry}$ between $\tilde{I}^{1:N}_1$ and $I^{1:N}_1$ using Eq. equation 6.
5:     Backward $\mathcal{L}_{render}$.
6: **until** SMGS converges
7: Initialize $\tilde{M}_t \leftarrow M_1, \hat{M} \leftarrow \emptyset$.
8: **for** each $Y_t$ in $Y_{1:T+1}$ **do**                    ▷ Extracting Mesh from Image Space
9:     **repeat**
10:         $\tilde{x}^W_{t+1} \leftarrow \tilde{x}^W_t + \tilde{\Delta} x^W_t$
11:         $\tilde{M}_{t+1} \leftarrow (\tilde{x}^W_{t+1}, \tilde{x}^M_{t+1}, E)$
12:         $\tilde{I}_{t+1} \leftarrow (SMGS)(\tilde{M}_{t+1}, View_{[1:N]})$
13:         Compute geometry loss $\mathcal{L}_{geometry}$ between $\tilde{I}^{1:N}_{t+1}$ and $I^{1:N}_{t+1}$ using Eq. equation 8.
14:         Backward $\mathcal{L}_{geometry}$.
15:     **until** $\Delta x^W_t$ converges
16:     $\tilde{x}^W_{t+1} \leftarrow \tilde{x}^W_t + \tilde{\Delta} x^W_t$.
17:     $\tilde{M}_{t+1} \leftarrow (\tilde{x}^W_{t+1}, \tilde{x}^M_{t+1}, E)$
18:     Append $\tilde{M}^W_{t+1}$ to $\hat{M}$.
19: **end for**
20: **for** each $\hat{M}_t$ in $\hat{M}_{1:T+1}$ **do**         ▷ Train Dynamics Simulator
21:     $\tilde{M}_{t+1} \leftarrow GNN(\hat{M}_t)$
22:     $(\tilde{x}^W_{t+1}, \tilde{x}^M_{t+1}, E) \leftarrow \tilde{M}_{t+1}$
23:     Compute node loss $\mathcal{L}_{node}$ between $\tilde{x}_{t+1}$ and $\hat{x}_{t+1}$ using Eq. equation 9.
24:     Backward $\mathcal{L}_{node}$.
25: **end for**
26: return 2D-3D mapping SMGS, $neural simulator$ GNN

---

**Algorithm S.2** The forward process of DVC in CloDS.

---

**Input:** Sequences of different viewpoints $View_n$, initial mesh state $M_0 = (x^W_0, x^M, E)$, neural dynamic simulator GNN and neural render SMGS.

1: Initialize $I \leftarrow \emptyset$.
2: **for** $t = 1 \rightarrow T$ **do**
3:     $M_t \leftarrow GNN(M_{t-1})$
4:     $I_t \leftarrow SMGS(M_t, View_n)$
5:     Append $I_t$ to $I$.
6: **end for**
7: return video $I$

---

Generally, we consider the first 300 time steps of the first 100 trajectories as the **training set**, and the last 100 time steps of the first 100 trajectories and all time steps of the remaining 20 trajectories as the **test set**. In different tasks, there are slight variations in the training and testing data.

In the Cloth Dynamics Grounding task, the training data for MGN* and MGN consist of meshes from the first 300 time steps of the 1-50 trajectories and the first 300 time steps of the 1-100 trajectories, respectively. The training data for CloDS* and CloDS** consists of videos from the first 300 time steps of the 1-50 trajectories and the first 300 time steps of the 1-100 trajectories, respectively. CloDS uses videos from the 51-100 trajectories and meshes information from the 1-50 trajectories as training data. During testing, meshes from the 51-100 trajectories and all time steps of the test set are used as the test data. Since the initial states of the 51-100 trajectories have been seen by the model during training in most models (Except for MGN*, they are referred to as the 'Viewed' trajectories, whereas the trajectories in the test set represent cloth poses and positions that the model has never seen, and thus are referred to as the 'Unviewed' trajectories.

When assessing the performance of SMGS, we use multi-view images from the first 200 time steps of 5 trajectories as both training and testing data. SMGS generates Gaussian component representations of the cloth from the first frame of each trajectory and then generates multi-view images for 200 time steps by changing the mesh. The results from the 5 trajectories are averaged to obtain the final experimental result in table 3.

When evaluating CloDS's performance on the forward of DVC, for the baseline, since it cannot utilize multi-view videos, we use fixed-view images from the first 300 time steps of 5 trajectories for training, with the following 100 time steps as the test data. For CloDS, we use the corresponding multi-view videos as traning and test data. The final experimental result in tabel 4 is the average from 5 trajectories.

## D.2 EVALUATION METRICS

For different tasks, we use various evaluation metrics. Metrics such as Peak Signal-to-Noise Ratio (PSNR), LPIPS, and SSIM are commonly used and will not be elaborated on further in this paper. In this section, we mainly focus on the Rollout RMSE in the Cloth Dynamics Grounding task and the RMSE metric in Video Prediction task.

In the Cloth Dynamics Grounding task, the model predicts the mesh at each time step by performing a rollout from the first frame's mesh input. Therefore, Rollout RMSE is used to evaluate whether the model has learned correct cloth dynamics. Rollout RMSE is computed as the Root Mean Squared Error (RMSE) of position in Lagrangian systems and momentum in Eulerian systems. It averages the error across all spatial coordinates, mesh nodes, time steps in each trajectory, and all trajectories in the test data. The formula for Rollout RMSE is as follows:

$$d_{t \leq T}^{\text{AVG}} = \frac{1}{N} \sum_{i=1}^{N} \frac{1}{T} \sum_{t=1}^{T} \sqrt{\frac{1}{K} \sum_{j=1}^{K} (\hat{x}_{t,j}^{W} - \tilde{x}_{t,j}^{W})}, \tag{S.2}$$

$$d_{t > T}^{\text{AVG}} = \frac{1}{N} \sum_{i=1}^{N} \frac{1}{400 - T} \sum_{t=T}^{400} \sqrt{\frac{1}{K} \sum_{j=1}^{K} (\hat{x}_{t,j}^{W} - \tilde{x}_{t,j}^{W})}, \tag{S.3}$$

where $\hat{x}_{j}^{W}$ represents the predicted mesh node world-space coordinates, $N$ is the number of trajectories and $K$ is the number of nodes. The specific steps for obtaining this are:

$$\hat{x}_{t}^{W} = \text{GNN}(\text{GNN}(....\text{GNN}(x_{0}^{W})...)). \tag{S.4}$$

Specifically, the rollout method predicts the first frame based on the ground truth, and the subsequent time step predictions are obtained recursively to derive the system trajectory.

Similar to the Rollout RMSE, the RMSE in Table 4 is calculated as the average RMSE between the predicted image $\hat{I}_t$ and the ground truth image $I_t$ across all time steps of the rollout. The formula is as follows:

$$\text{RMSE}_{\text{VP}} = \frac{1}{T} \sum_{t=1}^{T} \text{RMSE}(\hat{I}_t, I_t). \tag{S.5}$$

Table S.1: Performance of CDR under different noise conditions. "+ Gaussian noise" refers to adding Gaussian noise with 0 mean and 0.001 variance to each node. "+ Translation noise" adds Gaussian noise separately along X, Y, and Z axes and averages errors. "+ Scaling noise" applies a random scaling factor in [0.95, 1.05] to the cloth.

| Method | Viewed | | | | Unviewed | | | |
| | Interpolation | | Extrapolation | | Interpolation | | Extrapolation | |
| | $d_{t\leq300}^{\text{AVG}}$ | $d_{t=300}$ | $d_{t\leq300}^{\text{AVG}}$ | $d_{t=400}$ | $d_{t\leq300}^{\text{AVG}}$ | $d_{t=300}$ | $d_{t\leq300}^{\text{AVG}}$ | $d_{t=400}$ |
|---|---|---|---|---|---|---|---|---|
| CloDS* | 0.1313 | 0.1305 | 0.1312 | 0.1336 | 0.1437 | 0.1418 | 0.1381 | 0.1411 |
| | ± 0.046 | ± 0.039 | ± 0.037 | ± 0.048 | ± 0.029 | ± 0.045 | ± 0.046 | ± 0.048 |
| + Gaussian noise | 0.1362 | 0.1347 | 0.1359 | 0.1367 | 0.1481 | 0.1463 | 0.1402 | 0.1423 |
| | ± 0.063 | ± 0.080 | ± 0.065 | ± 0.097 | ± 0.084 | ± 0.062 | ± 0.075 | ± 0.056 |
| + Translation noise | 0.1353 | 0.1332 | 0.1363 | 0.1378 | 0.1453 | 0.1479 | 0.1397 | 0.1430 |
| | ± 0.086 | ± 0.074 | ± 0.098 | ± 0.060 | ± 0.079 | ± 0.092 | ± 0.077 | ± 0.071 |
| + Scaling noise | 0.1434 | 0.1455 | 0.1451 | 0.1474 | 0.1495 | 0.1471 | 0.1423 | 0.1437 |
| | ± 0.062 | ± 0.065 | ± 0.077 | ± 0.089 | ± 0.064 | ± 0.071 | ± 0.053 | ± 0.068 |

## D.3 BASELINE MODELS

To evaluate the performance of CloDS, we compared it with different baseline methods. The detailed descriptions of each model are presented below.

In Cloth Dynamics Grounding, unlike existing methods that learn cloth dynamics using mesh-based labels, CloDS can learn the underlying physical dynamics of cloth from sparse viewpoint videos without requiring the use of physical representations as labels. Therefore, a fair comparison with current models that use mesh-based labels is challenging. In the CDG task, we compare MGN trained with different data and training methods. The specific data splits are presented in Appendix D.1.

**MGN (Pfaff et al., 2020):** MGN is a mesh-based general method that can model various physical systems accurately and effectively. It has good generalization ability and can be extended during inference. It uses a message-passing neural network (MPNN) with an Encode-Process-Decode architecture to predict properties such as acceleration for dynamic systems, and then the mesh state is obtained through an integrator.

In the dynamic scene novel view synthesis task, we need to compare SMGS with relevant baselines. The descriptions of the related baselines are as follows:

**4DGS (Wu et al., 2024a):** A new explicit representation method is proposed, which simultaneously contains three-dimensional Gaussian distributions and four-dimensional neural voxels. A decomposition-based neural voxel encoding algorithm, based on HexPlane (Hexagonal Plane), is introduced to effectively construct Gaussian features from the four-dimensional neural voxels. This approach then applies a lightweight MLP to predict the Gaussian deformation at new time steps.

**MSTH (Wang et al., 2024a):** A method for efficiently reconstructing dynamic 3D scenes in both time and space. It decouple the representation of dynamic radiance fields into time-invariant 3D hash encoding and time-varying 4D hash encoding. By using uncertainty-guided masks as weights, the algorithm avoids the large number of hash collisions caused by the additional time dimension.

**M5D-GS (Hu et al., 2025b):** This method addresses 3D dynamic scene reconstruction for objects undergoing large deformations. It builds on a 3D-GS framework to achieve high-fidelity, real-time rendering of dynamic objects. The framework explicitly decouples object motion and deformation estimation and employs a coarse-to-fine matching strategy to initialize large motions.

To evaluate the quality of videos generated by CloDS in the DVC forward process, we need to compare CloDS with video prediction models. The descriptions of the related baselines are as follows:

**MAU (Chang et al., 2021):** MAU attempts to simulate temporal information in videos by expanding the temporal receptive field and aggregating it through an attention mechanism. The proposed module is used in the bottleneck part of the autoencoder, where spatial and temporal information is combined across different MAU layers.

**TAU (Tan et al., 2023):** TAU captures time evolution by decomposing temporal attention into intraframe static attention and inter-frame dynamic attention. It is argued that mean square error loss mainly focuses on intraframe differences, which motivates the introduction of a differential divergence regularization to address inter-frame variations. By keeping the spatial encoder and decoder simple with 2D convolutional neural networks, we intentionally implement the proposed TAU modules and surprisingly find that the resulting model achieves performance competitive with recurrent-based models.

**MMVP (Zhong et al., 2023):** MMVP is an end-to-end trainable two-stream pipeline. It uses motion matrices which helps the motion prediction become more focused, and efficiently reduces the information loss in appearance to represent appearance-agnostic motion patterns.

**SimVP (Gao et al., 2022) :** SimVP is a simpler yet more effective video prediction model. It is entirely CNN-based and trained end-to-end using MSE loss, without introducing any additional techniques or complex strategies.

### D.4 EXPERIMENT PLATFORM

We conduct comprehensive experiments utilizing the NVIDIA H800 80GB PCIe paired with the Intel(R) Xeon(R) Platinum 8380 CPU @ 2.30GHz.

### D.5 TRAIN DETAILS

For MGN, we used the same settings as in the original paper (Pfaff et al., 2020). When training the Gaussian components with multiple trajectories, the first time step of all training trajectories is used to train the Gaussian components. The experiments in this paper are conducted on H800 GPUs. One complete pass through the first time step of all trajectories is considered an epoch. In each epoch, we loop 100 times for each time step. During the training of the Dynamics Learning GNN, we train the model for for 1000 epochs. The quantitative experimental results in the paper are the averages over five trials. Our code is available at https://github.com/whynot-zyl/CloDS.

### D.6 INFERENCE TIME OF CLODS

Our method is built upon a neural simulator and a mesh-based Gaussian splatting render. The neural physical simulator accelerates traditional physics solvers. And Gaussian splatting, which renders scenes through Gaussian volume projection, is significantly faster than conventional renderer. The rendering speed (FPS) on single H800 is reported in Figure S.1. As shown, our pipeline is substantially more efficient than traditional approaches.

## E ROBUSTNESS TO INITIAL MESH ERRORS

Since the method in this paper requires using the mesh from the first time step of the trajectories for deploying the Gaussian components to obtain the cloth's Gaussian component representation, an important assumption of CloDS is that an estimate of the initial cloth mesh is available. However, estimating the initial mesh may introduce errors due to various factors, so we need to demonstrate the robustness of CloDS to the noise of initial mesh. To assess CloDS's robustness on CDG, we add different noise to the initial mesh and then use the SMGS rendered from the noisy initial mesh to train the CloDS*. Finally, we

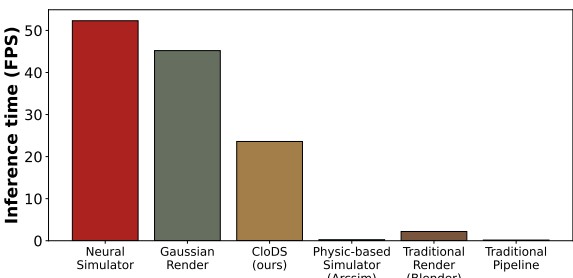

Figure S.1: Inference Time of CloDS (Neural Simulator + Gaussian Render) and tradition pipline (Physic-based Simulator + Traditional Render).

test it on the same test data as in Table 2. The results are presented in Table S.1. To more intuitively examine the model's inference behavior under noise perturbations, we visualize the corresponding results in Figure S.2. It can be observed that although errors in the initial mesh reduce CloDS's

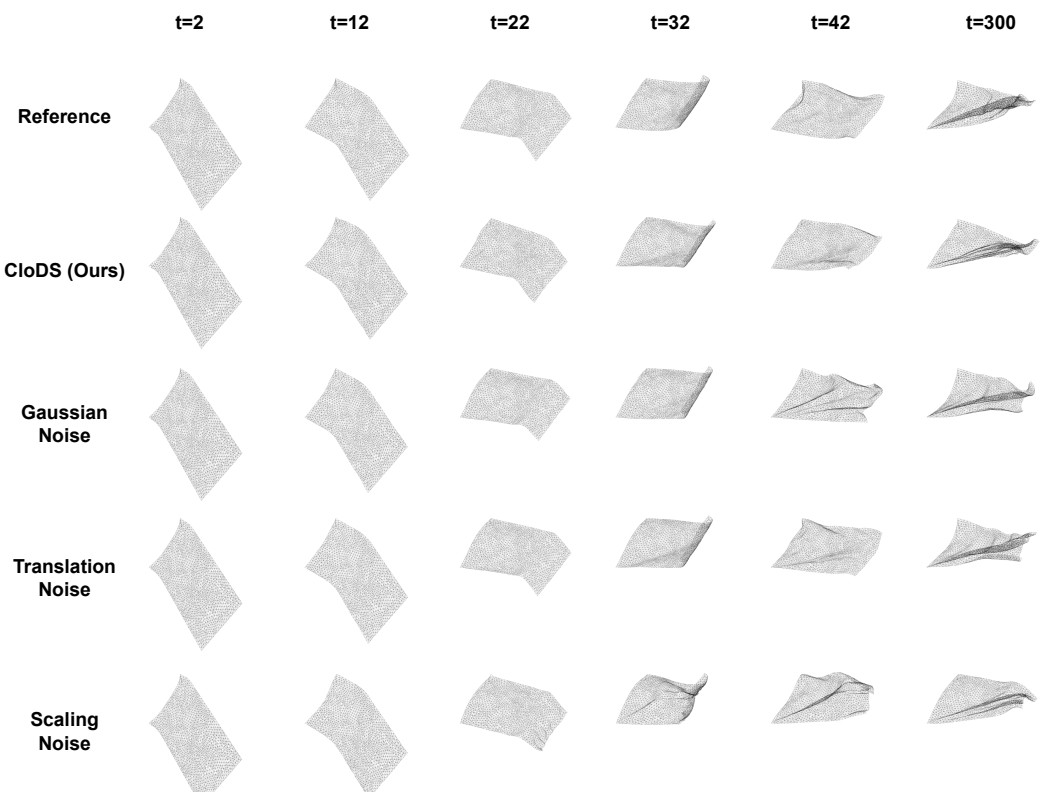

Figure S.2: Visualization of CDR under different noise conditions, corresponding to Table S.1. Videos are available in Part 11 at URL.

ability to learn cloth dynamics, CloDS still enables MGN to learn reliable cloth dynamics even in the presence of mesh errors.

The results show that after training with multiple trajectories, the video prediction model performs significantly worse on new cloth poses compared to CloDS. This is due to the strong deformations of the cloth, which exhibit chaotic behavior: small changes in the initial state can lead to drastically different subsequent states. Additionally, it is observed that after training on multiple trajectory videos, the video prediction model's extrapolation performance is worse than when trained on a single trajectory, while CloDS achieves better performance. This is because cloth dynamics are complex, and video prediction models struggle to effectively learn the dynamics of cloth. In contrast, CloDS maps 2D images to the 3D physical representation of the cloth, enabling it to better learn cloth dynamics from observation.

## F  RENDERED TEXTURES

To demonstrate the effectiveness and generalization ability of the CloDS, we rendered the cloth using different textures. All texture images used in this study are shown in Figure S.3. We first used the texture in Figure e S.3a to render multi-view videos, which produced the experimental results in Figure 5, Figure 6, and Table 2, showing that CloDS can learn cloth dynamics from multi-view videos in an unsupervised manner. We then used the texture in Figure S.3b to render a cylindrical cloth, demonstrating that our method generalizes to different cloth shapes (shown in Figure 7a). Finally, we used the texture in Figure S.3c to demonstrate that our method can adapt to various textures (shown in Figure 7b).

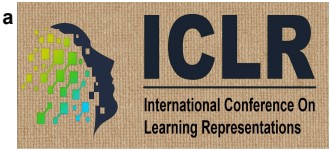 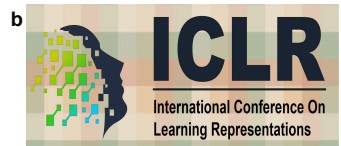 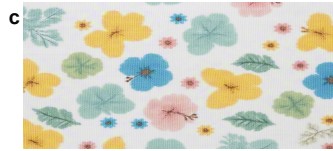

Figure S.3: **(a):** Texture used in Figure 6. **(b):** Texture used in Figure 7a. **(c):** Texture used in Figure 7b.

Table S.2: Performance on cloth dynamic video prediction. "Extrapolation" refers to testing the last 100 frames of the 10 training trajectories, while "Generalization" refers to testing the first 100 frames of 2 unseen trajectories.

| Method | Extrapolation | | | | Generalization | | | |
| --- | --- | --- | --- | --- | --- | --- | --- | --- |
| | PSNR (dB)↑ | SSIM↑ | LPIPS↓ | RMSE↓ | PSNR (dB)↑ | SSIM↑ | LPIPS↓ | RMSE↓ |
| MAU | 17.1273 | 0.9531 | 0.03743 | 0.13431 | 12.6789 | 0.9291 | 0.04817 | 0.16531 |
| TAU | 17.5812 | 0.9563 | 0.03472 | 0.13386 | 13.0322 | 0.9391 | 0.04532 | 0.16087 |
| MMVP | 17.5723 | 0.9523 | 0.03389 | 0.13549 | 12.9535 | 0.9380 | 0.04486 | 0.15791 |
| SimVP | 18.9722 | 0.9557 | 0.03246 | 0.12704 | 14.2391 | 0.9438 | 0.04593 | 0.14028 |
| CloDS (ours) | **27.7331** | **0.9835** | **0.00832** | **0.03987** | **25.2380** | **0.9798** | **0.01178** | **0.05901** |

## G    MORE RELATED WORK

**Video prediction model.**    Video prediction models learn object dynamics from videos to predict future states from initial frames (Gao et al., 2022; Tang et al., 2023; Zhong et al., 2023). These models are typically tested on datasets such as TaxiBJ (Zhang et al., 2017) and Moving MNIST (Srivastava et al., 2015) which focus on simple motion with minimal occlusion/deformation. In contrast, CDG demands 3D-aware modeling of highly deformable objects. In CloDS, we use a dynamics learning model to learn the cloth dynamics in 3D space and employ Gaussian Splatting Rendering as a mapping between 3D space and 2D pixel space. This allows the dynamic prediction model to learn 3D spatial information from the pixel space.

## H    MORE RESULT

### H.1    VIDEO PREDICTION MODEL TRAINED ON MULTIPLE TRAJECTORIES.

Currently, video prediction models can learn the motion patterns of objects from videos. However, these models typically focus on predicting rigid bodies, with less attention given to highly deformable objects like cloth. The results in the table 2 indicate that video prediction models are less effective than CloDS at learning the cloth dynamics by training on specific initial states. To further compare the generalization abilities of CloDS and video prediction models, we train video prediction models on the first 300 time steps of 10 trajectories and then test its interpolation performance on 2 unseen trajectories. Additionally, to assess the impact of increased training data on model performance, we test the extrapolation performance on the last 100 time steps of the 10 training trajectories. The experimental results are shown in Table S.2.

### H.2    ERROR ACCUMULATION IN MESH EXTRACTION.

When mapping the 2D pixel space to the 3D space, SMGS is used to recursively extract the cloth mesh for all time steps, allowing the Dynamic Learning GNN to learn the cloth dynamics. Since $\tilde{M}_{t+1}$ depends on $\tilde{M}_t$, errors can accumulate during the recursive extraction process. To explore the accumulation of errors, we calculated the error between the extracted mesh and the ground truth as it evolves over time steps, as shown in Figure S.4a, with the mean and variance presented in Figure S.4b. We can observe that as time progresses, the accumulated error does not increase dramatically but rather stabilizes. This indicates the stability and effectiveness of using SMGS for mesh extraction.

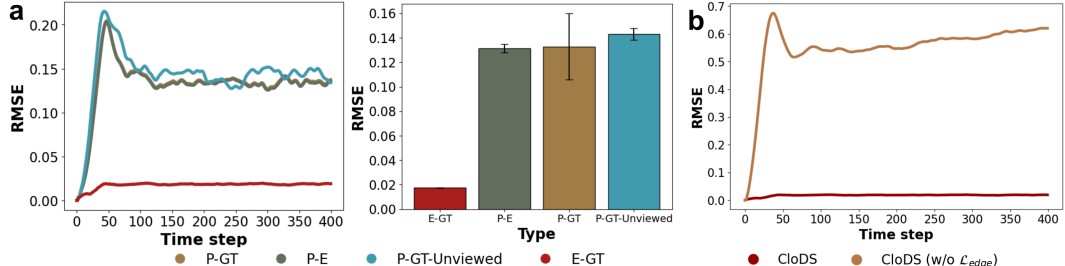

Figure S.4: **(a):** Error accumulation, mean and variance. "P-GT" is the error between the predicted and ground truth meshes. "P-E" is the error between the predicted and extracted meshes. "E-GT" is the error between the extracted and ground truth meshes. These are the experimental results on the viewed trajectories (1-50). "P-GT-UnViewed" is the error between the predicted and the extracted meshes on unviewed trajectories. **(b):** Error accumulation when with and without $\mathcal{L}_{edge}$.

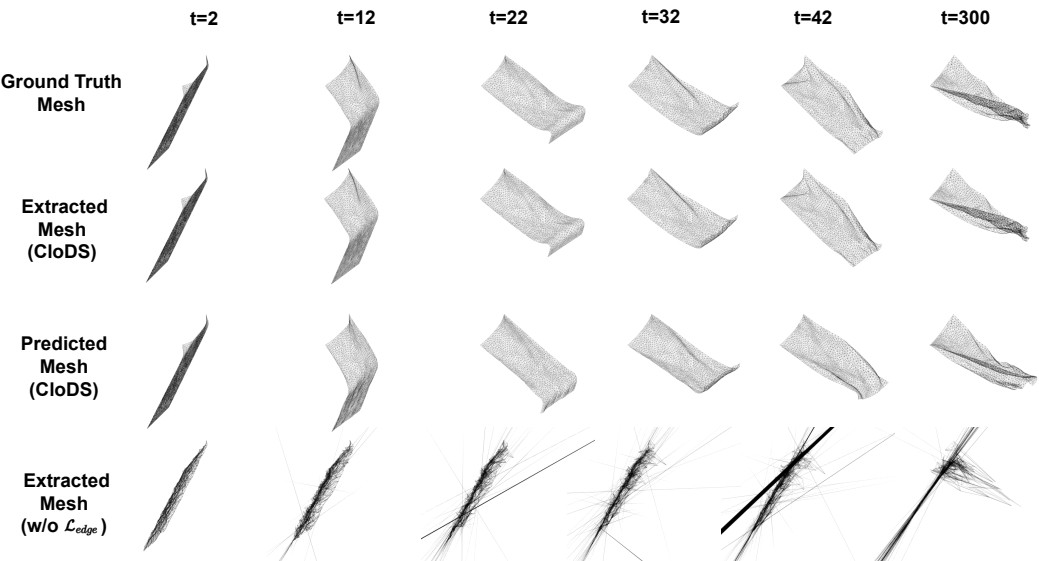

Figure S.5: The mesh visualizations corresponding to the results in Figure S.4. "Extracted mesh (CloDS)" and "Predicted mesh (CloDS)" correspond to the results presented in Figure S.4a, while "Extracted mesh (w/o $\mathcal{L}_{edge}$)" corresponds to the results shown in Figure S.4b. Videos are available in Part 12 at URL.

## H.3 ERROR ACCUMULATION IN DYNAMICAL SYSTEM PREDICTION.

When predicting the dynamics of the system, we use the rollout method to recursively predict the cloth state based on the initial state. Since the model does not learn the stationary state of the cloth, this simulation can lead to error accumulation, causing prediction errors to increase dramatically over time. Figure S.4a shows the error between the predicted mesh, the ground truth and the extracted mesh over time. For clearer visualization, we present the ground-truth mesh, the extracted mesh, and the predicted mesh in Figure S.5. We observe that even with full rollout, the error does not increase drastically, which confirms the stability of the cloth dynamics learned by CloDS.

Additionally, Figure S.4a shows that the mean error between the prediction and the ground truth, as well as the extracted mesh, are quite similar, indicating that the cloth dynamics learned by CloDS are close to the real cloth dynamics. However, the variance of the error between the prediction and the ground truth is larger than that between the prediction and the extracted mesh. This is because CloDS primarily uses the extracted mesh to enable the GNN to learn the cloth dynamics, leading to a more stable error between the prediction and the extracted mesh.

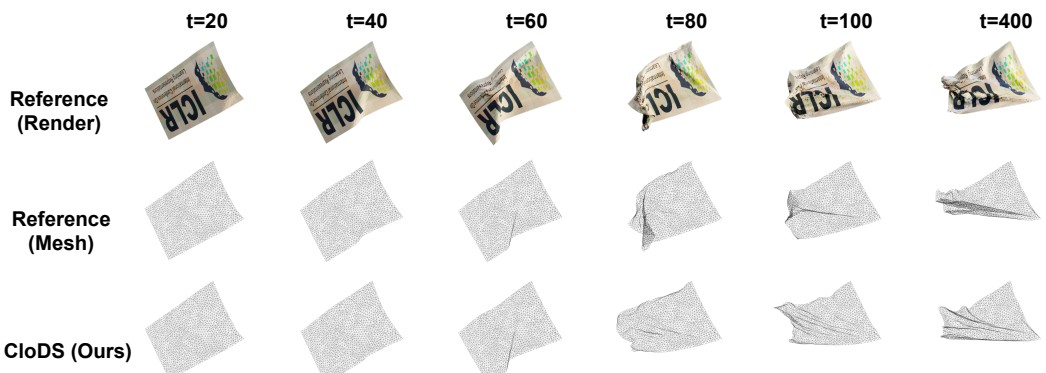

Figure S.6: Visualization results under lighting conditions. "Reference (Image)" shows the rendered images with lighting. "Reference (Mesh)" shows the corresponding meshes. "CloDS (ours)" shows the meshes predicted by CloDS. Videos are available in Part 6 at URL.

### H.4 NECESSITY OF MESH REPRESENTATION

In traditional simulators, particle-based representation is effective and well-suited for cloth simulation. However, in neural network (NN) methods, if gradient descent is used to randomly update particle positions without mesh-based constraints on their relative positions, NN will fail to converge. This is why this paper introduces a geometry loss based on mesh (Eq. 8). Without edge loss, the method degrades to a particle-based approach that ignores mesh topology. It can be seen that from the resulting error shown in Figure S.4b that the absence of mesh topology constraints leads to accumulation of node errors. To more clearly illustrate the necessity of the mesh representation, we visualize in Figure S.5 the mesh extracted without the edge loss (shown as "Extracted Mesh (w/o $\mathcal{L}_{edge}$)"). Moreover, by inspecting the point-cloud–based LIP results in Figure 9a, we observe that the cloth shape collapses rapidly during inference. This further corroborates the necessity of mesh-based representations for addressing the CDG problem.

### H.5 CLODS PERFORMANCE UNDER LIGHTING CONDITIONS.

To demonstrate that our method can also learn cloth dynamics under complex environmental conditions, we introduced lighting effects during rendering to generate multi-view video data with light and shadow variations. We then used CloDS to learn cloth dynamics under complex

Table S.3: Performance of CloDS in CDG under lighting and without lighting Conditions.

| Method | Interpolation | | Extrapolation | |
|---|---|---|---|---|
| | $d_{t\leq300}^{\text{AVG}}$ | $d_{t=300}$ | $d_{t\leq300}^{\text{AVG}}$ | $d_{t=300}$ |
| light | 0.1569±0.055 | 0.1582±0.044 | 0.1573±0.049 | 0.1541±0.052 |
| w/o light | 0.1313±0.046 | 0.1305±0.039 | 0.1312±0.037 | 0.1336±0.048 |

lighting conditions. The visual and quantitative results are shown in Figure S.6 and Table S.3. The generated videos are available in Part 6 at URL. The results indicate that even with lighting CloDS still learns cloth dynamics effectively. However, compared with the performance under no-light conditions, lighting reduces CloDS performance. This is mainly because the same region may exhibit different colors due to shadows and illumination, which disrupts the temporal consistency of the images. This inconsistency causes inaccurate mesh node estimation and reduces the accuracy of subsequent training. Although Table S.3 shows a performance drop under lighting compared to no-light conditions, CloDS still learns cloth dynamics effectively.

Moreover, CloDS significantly outperforms existing geometry-aware approaches. As shown in our response to Figure 9, current geometry-aware baselines already struggle to capture complex cloth geometry even under simple lighting conditions, and we sincerely thank you for this constructive suggestion on baseline construction.

Table S.4: Performance of CloDS on the CDR task under different camera number.

| Method | Interpolation | | Extrapolation | |
|---|---|---|---|---|
| | $d^{\text{AVG}}_{t\leq300}$ | $d_{t=300}$ | $d^{\text{AVG}}_{t\leq300}$ | $d_{t=300}$ |
| $n=10$ | 0.1362 | 0.1371 | 0.1401 | 0.1396 |
| | ± 0.082 | ± 0.077 | ± 0.081 | ± 0.069 |
| $n=20$ | 0.1347 | 0.1353 | 0.1334 | 0.1413 |
| | ± 0.073 | ± 0.065 | ± 0.082 | ± 0.066 |
| $n=30$ | 0.1313 | 0.1305 | 0.1312 | 0.1336 |
| | ± 0.046 | ± 0.039 | ± 0.037 | ± 0.048 |

Table S.5: Performance of CloDS on CDR task under different Gaussian component density.

| Method | Interpolation | | Extrapolation | |
|---|---|---|---|---|
| | $d^{\text{AVG}}_{t\leq300}$ | $d_{t=300}$ | $d^{\text{AVG}}_{t\leq300}$ | $d_{t=400}$ |
| $n=5$ | 0.1348 | 0.1389 | 0.1367 | 0.1387 |
| | ± 0.075 | ± 0.063 | ± 0.082 | ± 0.066 |
| $n=10$ | 0.1352 | 0.1368 | 0.1321 | 0.1423 |
| | ± 0.057 | ± 0.064 | ± 0.049 | ± 0.058 |
| $n=20$ | 0.1313 | 0.1305 | 0.1312 | 0.1336 |
| | ± 0.046 | ± 0.039 | ± 0.037 | ± 0.048 |

## H.6 THE IMPACT OF THE NUMBER OF CAMERAS.

For each trajectory, we set up 30 identical camera positions following NeuroFluid (Guan et al., 2022). We further explored the impact of number of cameras on Novel View Synthesis in the Dynamic Scenes task. As shown in Table S.4, more views improve performance by capturing finer details, despite gains from additional cameras are modest.

## H.7 THE IMPACT OF GAUSSIAN COMPONENT DENSITY

A higher density of Gaussian components allows for a more detailed representation of the 2D spatial features in the image. To assess the impact of Gaussian component density on model performance, we evaluate CloDS with varying densities and test its performance on the CDG task using the viewed trajectories. The experimental results are presented in Table S.5. It can be observed that increasing the Gaussian component density leads to a slight improvement in model performance. However, training with a lower density still enables the model to effectively learn the cloth dynamics.

## H.8 CLODS *vs.* SORA

As a representative of current generative models, Sora (Liu et al., 2024) can produce visually realistic videos by observing videos. However, its physical understanding is severely limited (Motamed et al., 2025). To evaluate Sora's performance on the CDG task, we provide the initial frame of a cloth and the prompt "A flag fluttering in the wind" to generate a cloth motion video. The generated videos are available in Part 7 at URL. It is evident that the results lack adherence to physical principles.

## H.9 EXTRACTED INITIAL MESH FROM THE INITIAL FRAME.

We use 2D gaussian splatting Huang et al. (2024) to extracte the Initial mesh $M_1$ from the first frames $I_1^{1:N}$. In general, the process consists of reconstructing a static field, rendering multi-view depth maps from this field, and finally applying TSDF fusion followed by Marching Cubes to obtain the initial triangular mesh. Concretely, we first use the first-frame images $I_1^{1:N}$ together with their camera intrinsics and extrinsics to obtain a Gaussian representation of the cloth via 2D Gaussian Splatting (2DGS). Unlike 3D Gaussian Splatting, 2DGS represents the surface using 2D elliptical disks embedded in 3D space. Each disk is parameterized by a 3D center, two tangent-plane directions, and the corresponding anisotropic scales. During forward rendering, 2DGS extends the CUDA renderer of 3DGS to additionally output per-pixel depth and surface normals, which are required for extracting the initial mesh $M_1$. Next, we perform a forward rendering pass for all training views, producing the depth and normal maps $D_1^{1:N}$. For each view $i$, we back-project every pixel into 3D space using the predicted depth $D_1^i$ and the known camera parameters, generating a multiview point cloud. We then define a voxel grid at a fixed resolution and construct a Truncated Signed Distance Function (TSDF) within this volume. All depth maps are fused into the TSDF using the built-in TSDF integration module of Open3D Zhou et al. (2018), resulting in a consistent volumetric reconstruction. Finally, applying Marching Cubes to the fused TSDF yields the initial mesh $M_1$. See Algorithm S.3 for algorithmic details.

---

**Algorithm S.3** Initial mesh extraction from first-frame NeRF-style data using 2D Gaussian Splatting

---

**Input:** Multi-view RGB images $I_1^{1:N}$ of the first frame; camera intrinsics and extrinsics $\{K_i, T_i\}_{i=1}^N$; voxel size $v$; truncation threshold $\tau$; number of training iterations $T$.
**Output** Initial triangle mesh $M_1$.

1: **(Optional) SfM / calibration point cloud.**
2: **if** a sparse calibration point cloud $\mathcal{P}$ is not provided **then**
3:     Run SfM on $\{I_1^{1:N}\}$ to obtain camera poses and sparse points
4:     $\mathcal{P} \leftarrow \{p_k\}_{k=1}^K$.
5: **else**
6:     Use the given $\mathcal{P}$ and camera poses.
7: **end if**

8: **Initialize 2D Gaussian splats.**
9: **for** each point $p_k \in \mathcal{P}$ **do**
10:     Set Gaussian center $c_k \leftarrow p_k$
11:     Initialize orthonormal tangential basis $(t_u^k, t_v^k, t_w^k)$ with $t_w^k = t_u^k \times t_v^k$
12:     Initialize scales $s_u^k, s_v^k \leftarrow s_{\text{init}}$
13:     Initialize opacity $\alpha_k \leftarrow \alpha_{\text{init}}$ and SH appearance coefficients $a_k$
14: **end for**
15: Let $\theta$ collect all Gaussian parameters $\{c_k, t_u^k, t_v^k, s_u^k, s_v^k, \alpha_k, a_k\}$.

16: **Train 2DGS on the first-frame views.**
17: **for** $t = 1$ to $T$ **do**
18:     **for** $i = 1$ to $N$ **do**                        ▷ for each training view
19:         Render color image $\hat{I}_1^i$ from $\theta$ using the 2DGS renderer
20:         Simultaneously render geometric buffers:
      depth samples $\{z_{ij}\}$ along each ray,
      median-depth map $\tilde{D}_1^i$,
      surface-normal map $N_1^i$
21:     **end for**
22:     Compute color reconstruction loss $L_c(\theta, I_1^{1:N}, \hat{I}_1^{1:N})$
23:     Compute depth distortion loss $L_d(\theta, \{z_{ij}\})$
24:     Compute normal consistency loss $L_n(\theta, \{\tilde{D}_1^i, N_1^i\})$
25:     $L \leftarrow L_c + \alpha L_d + \beta L_n$
26:     Update $\theta \leftarrow \theta - \eta \nabla_\theta L$ using Adam
27: **end for**
28: Denote the optimized parameters as $\theta^\star$.

29: **Render per-view depth maps from 2DGS.**
30: **for** $i = 1$ to $N$ **do**
31:     Using $\theta^\star$, render median-depth map $D_1^i$ for view $i$
32: **end for**

33: **TSDF fusion.**
34: Initialize a TSDF volume $V$ with voxel size $v$ and truncation $\tau$
35: **for** $i = 1$ to $N$ **do**
36:     **for** each pixel $x$ in $D_1^i$ **do**
37:         Back-project $(x, D_1^i(x))$ via $(K_i, T_i)$ to a 3D point $q$
38:         Update TSDF values and weights in $V$ for voxels within distance $\tau$ of $q$
39:     **end for**
40: **end for**

41: **Mesh extraction.**
42: Apply Marching Cubes to the zero-level set of $V$ to obtain a watertight mesh $M_1$
43: **return** $M_1$

---

