# CloDS: Visual-Only Unsupervised Cloth Dynamics Learning in Unknown Conditions

## Abstract

Deep learning has demonstrated remarkable capabilities in simulating complex dynamic systems. However, existing methods require known physical properties as supervision or inputs, and this dependence limits their applicability under unknown conditions. To explore this challenge, we introduce Cloth Dynamics Grounding (CDG), a novel scenario that involves unsupervised learning of cloth dynamics from multi-view visual observations. We further propose Cloth Dynamics Splatting (CloDS), an unsupervised dynamic learning framework designed for CDG. To enable unsupervised learning of cloth dynamics, we develop a three-stage training framework for CloDS. Moreover, to address the challenges posed by large non-linear deformations and severe self-occlusions in CDG, we introduce a dual-position opacity modulation that supports bidirectional mapping between 2D observations and 3D geometry via mesh-based Gaussian splatting. It jointly considers the absolute and relative position of Gaussian components. Comprehensive experimental evaluations demonstrate that CloDS effectively learns cloth dynamics from visual data while maintaining strong generalization capabilities for unseen configurations. Our code is available at `https://anonymous.4open.science/r/CloDS_ICLR/`. Visualization results are available at `https://anonymous.4open.science/r/CloDS_video_ICLR/`.

## 1 Introducion

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

$$\mathbf{r}_1 = \frac{(X_{t,2}^W - X_{t,1}^W) \times (X_{t,3}^W - X_{t,1}^W)}{\|(X_{t,2}^W - X_{t,1}^W) \times (X_{t,3}^W - X_{t,1}^W)\|}, \tag{4}$$

where $\times$ is the cross product. The second vertex is $\mathbf{r}_2 = \frac{(X_{t,2}^W - X_{t,1}^W)}{\|X_{t,2}^W - X_{t,1}^W\|}$. The third is obtained as a single step in the Gram–Schmidt process Strang (2000):

$$\mathbf{r}_3 = \frac{\text{orth}(X_{t,3}^W - X_{t,1}^W, r_1, r_2)}{\|\text{orth}(X_{t,3}^W - X_{t,1}^W, r_1, r_2)\|}. \tag{5}$$

The scale $S = \mathrm{diag}(s_1, s_2, s_3)$, where $s_1 = \epsilon$, $s_2 = \|X_{t,2}^W - X_{t,1}^W\|$ and $s_3 = \langle X_{t,3}^W - X_{t,1}^W, r_3 \rangle$. However, in CDG, strong self-occlusions and large deformations cause existing mesh-based GS methods to suffer from perspective distortions and color errors (Figure 3c). To address this, we modulate the Gaus-

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

|---|---|---|---|
| 3DGS | 39.6263 | 0.9986 | 0.00253 |
| 4DGS | 23.2089 | 0.9718 | 0.01582 |
| MSTH | 23.1353 | 0.9682 | 0.01653 |
| M5D-GS | 29.3428 | 0.9731 | 0.01297 |
| GaMeS | 33.0249 | 0.9937 | 0.00521 |
| SMGS (Ours) | **36.2368** | **0.9959** | **0.00353** |

nents overlap in occluded cloth areas. This results in perspective and rendering errors (Figure 3c). In contrast, SMGS allocates opacity based on both 3D world-space and mesh-space coordinates: the former preserves relative positioning to prevent perspective errors, while the latter constrains opacity to avoid transparency in unseen regions (Further ablation study on SMGS detailed in Section 5.6).

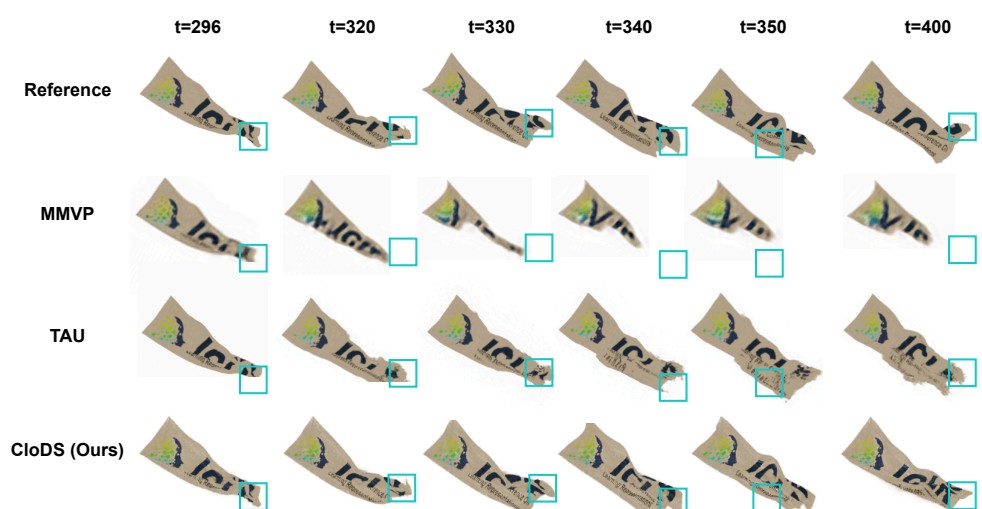

Figure 5: Visualization results of the DVC forward process and the video prediction model. Videos are available in Part 1 at URL.

## 5.4 CLODS OUTPERFORMS ON THE FORWARD PROCES OF DVC

Video prediction models learn physical dynamics from videos by generating future frames from current observations without reasoning about 3D structures. CloDS similarly uses video data for training. As shown in Figure 2b, after training CloDS predicts mesh states via the GNN and renders cloth-motion videos through SMGS. Table 4 com-

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

_{t\leq 300}^{\text{AVG}}$ | $d_{t=300}$ | $d_{t\leq 300}^{\text{AVG}}$ | $d_{t=400}$ |
| $n=5$ | 0.1348 | 0.1389 | 0.1367 | 0.1387 |
| $n=10$ | 0.1352 | 0.1368 | 0.1321 | 0.1423 |
| $n=20$ | 0.1313 | 0.1305 | 0.1312 | 0.1336 |

the accuracy of subsequent training. Although Table S.3 shows a performance drop under lighting compared to no-light conditions, CloDS still learns cloth dynamics effectively.

### H.6 THE IMPACT OF THE NUMBER OF CAMERAS.

For each trajectory, we set up 30 identical camera positions following NeuroFluid Guan et al. (2022). We further explored the impact of number of cameras on Novel View Synthesis in the Dynamic Scenes task. As shown in Table S.4, more views improve performance by capturing finer details, despite gains from additional cameras are modest.

### H.7 THE IMPACT OF GAUSSIAN COMPONENT DENSITY

A higher density of Gaussian components allows for a more detailed representation of the 2D spatial features in the image. To assess the impact of Gaussian component density on model performance, we evaluate CloDS with varying densities and test its performance on the CDG task using the viewed trajectories. The experimental results are presented in Table S.5. It can be observed that increasing the Gaussian component density leads to a slight improvement in model performance. However, training with a lower density still enables the model to effectively learn the cloth dynamics.

### H.8 CLODS vs. SORA

As a representative of current generative models, Sora Liu et al. (2024) can produce visually realistic videos by observing videos. However, its physical understanding is severely limited Motamed et al. (2025). To evaluate Sora's performance on the CDG task, we provide the initial frame of a cloth and the prompt "A flag fluttering in the wind" to generate a cloth motion video. The generated videos are available in Part 7 at URL. It is evident that the results lack adherence to physical principles.

## I ETHICS STATEMENT

We confirm that all authors have read and adhered to the ICLR Code of Ethics. This work complies with the ethical principles outlined therein, and no part of this study violates ethical standards.

## J REPRODUCIBILITY STATEMENT

We have made every effort to ensure the reproducibility of our results. A complete description of our model architecture, training procedure and evaluation metrics is our paper, with additional implementation details in Appendix D. Our anonymized source code, including data preprocessing scripts and configuration files, is available at `https://anonymous.4open.science/r/CloDS_ICLR/`. This repository enables the reproduction of all experiments reported in the paper.

## K THE USE OF LARGE LANGUAGE MODELS

In preparing this manuscript, we used a Large Language Model (ChatGPT, GPT-5) solely for language refinement and grammar editing. All research ideas, experimental designs, analyses, and interpretations were conceived and executed by the authors. The model did not contribute to research ideation, data collection, analysis, or the generation of novel scientific content. All scientific claims, interpretations, and conclusions are entirely our own, and the authors take full responsibility for the contents of this paper.