# OpenReview forum: "CloDS: Visual-Only Unsupervised Cloth Dynamics Learning in Unknown Conditions"
_ICLR.cc/2026/Conference — ICLR 2026 Poster_

### Official Review · Reviewer_FVhY · 2025-10-29

**Soundness:** 3
**Presentation:** 3
**Contribution:** 3
**Rating:** 4
**Confidence:** 5

**Summary:**

This paper introduces Cloth Dynamics Splatting (CloDS), an unsupervised, visual-only framework for Cloth Dynamics Grounding, which aims to learn cloth dynamics from multi-view videos under unknown conditions without direct physical supervision. The core strength lies in its ability to bridge the gap between 2D visual observations and 3D physical representations for highly deformable materials, a significant challenge in the field.

**Strengths:**

- The paper clearly defines a new and challenging problem, Cloth Dynamics Grounding, which focuses on unsupervised learning of cloth dynamics solely from visual data.

- The introduction of Spatial Mapping Gaussian Splatting, a mesh-based Gaussian splatting module, provides a differentiable mapping between 2D pixel space and 3D geometry. The proposed dual-position opacity modulation in SMGS is a clever solution to address severe self-occlusions and large non-linear deformations inherent to cloth dynamics. For the mesh-based gaussian splatting, there are relative work that should be cited.

[1]SuGaR: Surface-Aligned Gaussian Splatting for Efficient 3D Mesh Reconstruction and High-Quality Mesh Rendering

[2]real-time large-scale deformation of gaussian splatting

[3]VR-GS: A Physical Dynamics-Aware Interactive Gaussian Splatting System in Virtual Reality

[4]Recent Advances in 3D Gaussian Splatting

- CloDS achieves performance close to fully mesh-supervised methods on the CDG task, demonstrating effective unsupervised dynamics learning.

**Weaknesses:**

- The method assumes an initial mesh state ($M_1$) is available to build the initial Gaussian component representation. Although robustness to initial mesh errors is analyzed, I still suggest that some visual results should be prepared and presented to incorporate the results reported in FigureS.2.

- Performance degrades under complex lighting conditions due to temporal inconsistency caused by shadows and illumination, suggesting the current approach is sensitive to visual changes beyond pure geometry and dynamics.

- How is the initial mesh $M_1$ "extracted from the initial frame $I_1^{1:N}$ via 2D Gaussian Splatting"? Is this step fully unsupervised, or does it rely on any pre-trained model, shape priors, or a fixed template mesh? A brief explanation of how $M_1$ is obtained would clarify the unsupervised visual-only premise.

- The comparison to video prediction models is strong, but since a key challenge is 3D-aware modeling, a comparison to other geometry-aware, unsupervised methods (e.g., scene flow-based approaches or other particle-based visual grounding methods like NeuroFluid, adapted for cloth) would further solidify the value of the DVC and SMGS approach for this task.

- DeepFashion3D is also an impressive cloth dataset, some evaluations on the dataset are more encouraged.

**Questions:**

See weaknesses

---

> ### Author Response · Authors · 2025-11-21
> **Rebuttal by Authors (1/2)**
>
> We sincerely thank you for the constructive comments and suggestions, which are very helpful in improving our paper. The dataset and baseline you suggested are highly valuable for strengthening our paper. We have updated the paper according to reviews' suggestion, and all revisions related to your comments are highlighted in **light blue**. The visualization videos referenced in the rebuttal are available at ***https://anonymous.4open.science/r/ICLR_rebuttal_video***. Please do feel free to let us know if you have any further questions or guidance.😊
>
>
> >**Q1. Relative work that should be cited.**
>
> **Replay：** Excellent comment! We have added the paper you suggested to the Related Work section, and the corresponding modifications are highlighted in light blue.
>
> >**Q2. Visualization of Experimental Results for “Robustness to Initial Mesh Errors” (Table S.1) and Figure S.4 (formerly Figure S.2).**
>
> **Replay：** Thanks for your valuable feedback! It greatly contributes to improving the readability of our paper.
> In response to your comment, we have made the following revisions, and the corresponding changes are highlighted in light blue in the manuscript.
>
> - We have added **Figure S.2** in Appendix E. Figure S.2 visualizes the predictions of CloDS under different initial mesh errors (Table S.1). The corresponding videos are provided in Part 4 at ***https://anonymous.4open.science/r/ICLR_rebuttal_video***. We can observe that CloDS is able to produce plausible dynamics even under varying initial mesh distortions, further demonstrating the robustness of our approach.
>
> - We have added **Figure S.5** corresponding to Figure S.4 (formerly Figure S.2). Figure S.5 presents the ground-truth mesh, the mesh extracted by CloDS, the CloDS-predicted mesh, and the mesh extracted without enforcing mesh-connectivity constraints. The corresponding videos are provided in Part 5 at:  ***https://anonymous.4open.science/r/ICLR_rebuttal_video***. We can observe that the mesh extracted by CloDS closely matches the ground truth, whereas removing mesh-structure constraints leads to rapid error accumulation in the extracted mesh.
>
> >**Q3. Sensitive to visual changes beyond pure geometry and dynamics.**
>
> **Replay：** Great remark！As you correctly pointed out, the performance of CloDS has  **lighting reduce** under complex lighting conditions. This is also one of the key conclusions drawn from our experiments in Appendix H.5. However, we would like to clarify that although Table S.3 shows a slight performance drop under complex lighting conditions, this minor degradation indicates that **CloDS still effectively learns accurate cloth dynamics**. The visualizations provided in Figure S.6 further support this observation. In other words, CloDS is able to robustly learn both the geometry and the underlying dynamics even under varying lighting conditions, with visual changes causing only a slight reduction in performance.
>
> Moreover, CloDS significantly outperforms existing geometry-aware approaches. As shown in our response to Q6, current geometry-aware baselines already **struggle to capture complex cloth geometry even under simple lighting conditions**, and we sincerely thank you for this constructive suggestion on baseline construction.

---

> ### Author Response · Authors · 2025-11-21
> **Rebuttal by Authors (2/2)**
>
> >**Q4. How is the initial mesh "extracted from the initial frame via 2D Gaussian Splatting"?**
>
> **Replay：** Excellent question! Briefly, we first apply 2D Gaussian splatting to obtain the depth maps $D_1^{1:N}$ for the first frame from all camera views. We then use TSDF fusion and Marching Cubes implemented in Open3D [1] to reconstruct the initial mesh $M_1$ from these depth maps. The detailed procedure and algorithmic table have been added to **Appendix H.9** and highlighted in light blue.
>
>
> >**Q5. Does "extracted from the initial frame" rely on any pre-trained model?**
>
> **Replay：** Insightful question! Thanks for your careful reading. In fact, the extraction of the initial mesh **does not require any pre-trained model**. As discussed in Q4, 2D Gaussian splatting enables the recovery of depth information directly from images, which can then be used for downstream mesh construction. Therefore, initial mesh extraction can be performed without any pre-trained model. A similar strategy is also employed in Gaussian Garments[2], referenced by Reviewer kW74.
>
>
>
> >**Q6. Comparison to geometry-aware, unsupervised methods.**
>
> **Replay：** Thanks for this insightful comment! This suggestion is extremely valuable for improving our manuscript. To address your concern, we adapted the existing methods LIP [3] and CS [4] to the CDR task. LIP uses **point clouds** as the geometric representation, while CS models cloth using **meshes**. It is important to note that although LIP and CS are geometry-aware, they are **not fully unsupervised**: both rely on pretrained models, in contrast to our fully unsupervised CloDS setting.
>
> To adapt these methods to the cloth dynamic grounding settings, we pretrain the Particle Posterior Estimator and the Probabilistic Particle Simulator in LIP on cloth point-cloud data. For CS, we pretrain the GNN on cloth-mesh data.
>
> The predicted 3D geometry is visualized in **Figure 9a**. Videos are available in Part 1 at ***https://anonymous.4open.science/r/ICLR_rebuttal_video***. The quantitative results are presented in **Table A**.
>
> **Table A.** Average RMSE between predicted mesh nodes and ground truth.
>
> |Model|RMSE|
> |:-:|:-:|
> |**LIP**|1.975 $\pm$ 0.538|
> |**CS**|0.7402 $\pm$ 0.098|
> |**CloDS(ours)**|**0.1302 $\pm$ 0.025**|
>
> We observe that LIP, which uses point clouds as its geometric representation, suffers from **rapidly accumulating errors** during rollout, leading to severe distortions and eventual collapse of the cloth shape. In contrast, CS represents cloth with meshes, and the mesh connectivity provides additional constraints that help preserve the cloth shape better than LIP. Both CS and LIP perform substantially worse than CloDS, which further demonstrates the effectiveness of our approach. The newly added analysis and results have been highlighted in light blue in the revised manuscript.
>
>
> >**Q7. Result on DeepFashion3D V2 (mesh).**
>
> **Replay：** Excellent suggestion, and thank you for directing us to such a high-quality dataset! We construct a Real-Garment training and test dataset by performing physics-based simulations on high-quality garment meshes from the DeepFashion3D V2 dataset. CloDS is then trained and evaluated on this dataset, with the quantitative and qualitative results shown in **Figure 9c**. Videos are available in Part 3 at ***https://anonymous.4open.science/r/ICLR_rebuttal_video***. The quantitative results are presented in **Table B**. We observe that CloDS learns reliable dynamics even on realistic garment geometries, indicating strong generalization to complex real-world shapes.
>
>
> **Table B.** Average RMSE of CloDS between predicted mesh nodes and ground truth in DeepFashion3D V2 dataset.
>
> |Dataset|RMSE|
> |:-:|:-:|
> |**Pants**|0.065 $\pm$ 0.002|
> |**Dress**|0.016 $\pm$ 0.007|
>
> **Concluding remark:** We sincerely thank you for putting forward excellent comments. We hope the above responses are helpful to clarify your questions. We look forward to addressing any additional questions. Your consideration of improving the rating of our paper will be much appreciated!
>
> **References:**
>
> [1] Open3D: A modern library for 3D data processing
>
> [2] Gaussian Garments: Reconstructing Simulation-Ready Clothing with Photorealistic Appearance from Multi-View Video. 3DV 2025.
>
> [3] Latent Intuitive Physics: Learning to Transfer Hidden Physics from A 3D Video, ICLR 2024
>
> [4] Cloth-splatting: 3d cloth state estimation from rgb supervision, CoRL 2024

---

> > ### Comment · Reviewer_FVhY · 2025-11-22
> >
> > Thanks for your detailed response, the response has addressed my major concerns and I will raise the score for weak accept under the conditions: the mentioned experiments and explanations in the rebuttal should be added into the revised version of the paper.

---

> ### Author Response · Authors · 2025-11-22
> **Thank you for raising the score**
>
> Dear Reviewer FVhY,
>
> We greatly appreciate your positive feedback. The discussion with you has been quite productive and fruitful. Thank you very much for raising the score.
> In response to your questions and suggestions, we have made the following revisions to our paper, with all corresponding changes highlighted in light blue：
>
> >**Q1. Relative work that should be cite.**
>
> We have added the paper you suggested to the ***Related Work*** section. Thank you for your recommendation!
>
> >**Q2. Visualization of Experimental Results for “Robustness to Initial Mesh Errors” (Table S.1) and Figure S.4 (formerly Figure S.2).**
>
> We have added ***Figure S.2*** along with its explanations in ***Appendix E***, and ***Figure S.5*** with corresponding explanations in ***Appendix H.3***. Your suggestions substantially contribute to improving the quality of our paper.
>
> >**Q3. Sensitive to visual changes beyond pure geometry and dynamics.**
>
> Following your suggestion, we further emphasized the performance comparison between CloDS and other geometry-aware baselines in ***Appendix H.5***. We greatly appreciate your constructive recommendation regarding the baseline.
>
> >**Q4 & Q5. How is the initial mesh "extracted from the initial frame via 2D Gaussian Splatting"? & Does "extracted from the initial frame" rely on any pre-trained model?**
>
> We added a brief explanation of the extraction method in ***Appendix H.9*** and clarified that the extraction process does not require any pre-trained model.
>
> >**Q6.Comparison to geometry-aware, unsupervised methods.**
>
> This suggestion plays a crucial role in improving the quality of our paper. We sincerely appreciate your guidance on constructing the baseline. The corresponding updates have been incorporated into ***Section 5.6***.
>
> >**Q7. Result on DeepFashion3D V2 (mesh).**
>
> This high-quality dataset greatly enriches the presentation of our results. We have updated the corresponding content in **Section 5.7**.
>
> Thanks for your time and effort, and once again, we sincerely appreciate your decision to raise the rating of our paper.
>
> Best regards,
>
> The Authors

---

### Official Review · Reviewer_kW74 · 2025-11-01

**Soundness:** 2
**Presentation:** 1
**Contribution:** 2
**Rating:** 4
**Confidence:** 4

**Summary:**

This paper learns cloth dynamics from video. The approach first reconstructs a mesh from images via Gaussian splatting, then applies a mesh-based neural simulator to model dynamics in the mesh domain. Experiments show performance comparable to or better than selected baselines.

**Strengths:**

* The method learns cloth dynamics directly from video.
* The method achieves performance comparable to approaches trained on ground-truth mesh data.

**Weaknesses:**

1. Clarity and consistency. The writing is unnecessarily complex. If I understand correctly, a simple and clear description would be: learn dynamics directly from videos by first performing video-to-geometry grounding, then training a dynamics model on the grounded meshes. Also, there appears to be a typo/inconsistency: Equations (7) and (9) for geometry should take the same input parameters; please verify and correct.
2. Related work coverage (missing citations). Given the focus on data-driven, mesh-based cloth simulation, the Related Work should include additional existing work (e.g., [1,2,3]). In particular, [3] also learns cloth dynamics from multi-view videos and addresses more complex scenarios (richer appearance, human body interactions). Please discuss [3] in more detail and, if feasible, compare against the pipeline in [3].
3. Dataset scope and realism. The current dataset appears limited (a single cloth and ~120 videos of trajectories). Compared to [3], this setting may be simplistic. Please either expand the number/diversity of training and test videos or report results on established real-world datasets (e.g., those used in [3]) to demonstrate robustness and generality.
4. Experimental protocol and reporting. Since results are reported over 20 trajectories, include mean $\pm$ std across trajectories to reflect variability. Additionally, please report inference time (e.g., FPS/latency on a specified GPU) to quantify simulator efficiency.


[1]. Santesteban, et al. Self-Supervised Collision Handling via Generative 3D Garment Models for Virtual Try-On. CVPR 2021
[2]. Shao, et al. Towards Multi-Layered 3D Garments Animation. ICCV 2023.
[3]. Rong, et al. Gaussian Garments: Reconstructing Simulation-Ready Clothing with Photorealistic Appearance from Multi-View Video. 3DV 2025.

**Questions:**

Please refer to the weaknesses.

---

> ### Author Response · Authors · 2025-11-21
> **Rebuttal by Authors (1/2)**
>
> We sincerely thank you for the constructive comments and suggestions, which are very helpful in improving our paper. In particular, your guidance on our writing and experimental design has been profoundly beneficial to us. We have updated the paper according to reviews' suggestion, and all revisions related to your comments are highlighted in **light green**. The visualization videos referenced in the rebuttal are available at ***https://anonymous.4open.science/r/ICLR_rebuttal_video***. Please do feel free to let us know if you have any further questions or guidance.😊
>
> >**Q1. Clarity and consistency: The writing is unnecessarily complex.**
>
> **Replay：** We greatly appreciate this remarkable suggestion, which has been highly beneficial to improving our manuscript. Your understanding is absolutely correct. Thanks for your careful reading. Following your advice, we have revised portions of the Abstract, Introduction, and Method sections. We also added **Figure 2** in the Introduction to provide a concise overview of Differentiable Visual Computing, thereby improving the clarity of our paper.
>
> All related modifications are highlighted in light green. We hope these updates enhance the readability of our paper. If you notice anything that could be further improved, please feel free to point it out and we sincerely welcome your guidance.
>
> >**Q2. Clarity and consistency: The typo in Equations (7).**
>
> **Replay：** Thank you very much for your careful reading! Equation 5 (formerly Equation 7) was indeed missing the 3D spatial coordinates of initial mesh $x^W_{0}$. We have corrected this in the revised manuscript, and the modification is highlighted in light green.
>
> >**Q3. Related work coverage (missing citations).**
>
> **Replay：** Great comment! We have added the paper you suggested to the Related Work section, and the corresponding modification is highlighted in light green.
>
> >**Q4. Discussion of Gaussian Garments.**
>
> **Replay：** Insightful remark！Gaussian Garments is an excellent work. It reconstructs real garment geometry from multiview videos and enables downstream simulation. The method further fine-tunes a **pre-trained GNN** to recover the dynamic behavior of the reconstructed garments.
>
> At first glance, CloDS and Gaussian Garments appear similar: both reconstruct cloth geometry and subsequently train a GNN-based simulator. However, we would like to clarify that our work addresses an upstream problem relative to Gaussian Garments: ***How to obtain the pre-trained GNN in the first place?***
>
> Specifically, existing approaches acquire a pre-trained GNN through supervised learning on node-level trajectories, whereas **our goal is to obtain a pre-trained GNN in an unsupervised manner under unknown environments**. Therefore, **our method is fully compatible with Gaussian Garments** and can serve as its upstream module on the Gaussian Garments dataset.
>
> Unfortunately, the Gaussian Garments dataset has **not been released**, and we have currently been unable to obtain access to the ActorsHQ dataset[1] mentioned in the authors' GitHub repository. As a result, we are presently unable to demonstrate the combined performance of our method with Gaussian Garments.
>
> >**Q5. Dataset scope and realism: Diversity of training and test videos to demonstrate robustness and generality (real-world Garments).**
>
> **Replay：** Great comment! To compensate for the inability to integrate our method with Gaussian Garments in "Q4", and in response to the requests from both you and Reviewer FVhY, we construct a Real-Garment dataset by performing physics-based simulations on high-quality garment meshes from the DeepFashion3D V2 dataset [2]. The visualization of CloDS on this dataset are shown in **Figure 9c**, and the corresponding videos are provided in Part 3 at: ***https://anonymous.4open.science/r/ICLR_rebuttal_video***. The quantitative results are presented in **Table A**. As illustrated, CloDS successfully learns the dynamics of real garments, demonstrating the robustness and generality of our approach.
>
> **Table A.** Average RMSE of CloDS between predicted mesh nodes and ground truth in Real-Garment dataset.
>
> |Dataset|RMSE|
> |:-:|:-:|
> |**Pants**|0.065 $\pm$ 0.002|
> |**Dress**|0.016 $\pm$ 0.007|
>
> We have incorporated the relevant content into the Section 5.7, with modifications highlighted in light green and light blue (corresponding to Reviewer FVhY’s comments).
>
> **References:**
>
> [1] HumanRF: High-Fidelity Neural Radiance Fields for Humans in Motion, TOG 2023
>
> [2] Deep Fashion3D: A Dataset and Benchmark for 3D Garment Reconstruction from Single Images, ECCV 2020

---

> ### Author Response · Authors · 2025-11-21
> **Rebuttal by Authors (2/2)**
>
> >**Q6. Experimental protocol and reporting: Report "mean ± std".**
>
> **Replay：** Thanks for your excellent suggestion! We have added the corresponding standard deviations in Table 2, Table 3, Table 4, Appendix Table S.1, Table S.2, Table S.4, and Table S.5 in the main manuscript. The changes in response to this suggestion are highlighted in light green.
>
>
> >**Q7. Experimental protocol and reporting: Report inference time.**
>
> **Replay：** Great remark! We report the inference time (FPS) in **Appendix D.6**, and the corresponding updates are highlighted in light green.
>
> Specifically，Our method is built upon a neural simulator and a mesh-based Gaussian splatting renderer. The neural physical simulator accelerates traditional physic-based simulator. And Gaussian splatting, which renders scenes through Gaussian volume projection, is significantly faster than conventional renderer. The inference time (FPS) on single H800  is reported in **Table B**. We can observe that CloDS is substantially more efficient than traditional approaches (Traditional Pipeline).
>
>
> **Table B.** Inference time of CloDS (FPS).
>
> |Model|Dynamic Learner|Gaussian Render|CloDS(ours)|Physic-based Simulator (Arcsim)| Traditional Renderer (Blender)| Traditional Pipeline |
> |:-:|:-:|:-:|:-:|:-:|:-:|:-:|
> |**Inference time (FPS)**|**52.3**|**45.2**|**23.6**|0.24|2.2|0.16|
>
>
> **Concluding remark:** We sincerely thank you for reviewing our paper and putting forward thoughtful comments/suggestions. We hope the above responses are helpful to clarify your questions. We will be happy to hear your feedback and look forward to addressing any additional questions. Your consideration of improving the rating of our paper will be much appreciated!

---

> ### Author Response · Authors · 2025-11-26
> **Kindly request your feedback before the end of the discussion period**
>
> Dear Reviewer kW74:
>
> As the author-reviewer discussion period is soon ending, we would appreciate it if you could review our responses and provide your feedback at your earliest convenience. If there are any further questions or comments, we will do our best to address them before the discussion period ends.
>
> Thank you very much for your time and efforts! 😊
>
> Sincerely,
>
> The Authors

---

> ### Author Response · Authors · 2025-11-27
> **Kindly request your feedback before the end of the discussion period**
>
> Dear Reviewer kW74,
>
> I hope this message finds you well. As the discussion period is nearing its end, l wanted to ensure we have addressed all your concerns satisfactorily. If there are any additional points or feedback you'd like us to consider, please let us know. Your insights are invaluable to us, and we're eager to address any remaining issues to improve our work.
>
> Thank you for your time and effort in reviewing our paper! 😊
>
>
> Best regards,
>
> The Authors

---

> ### Author Response · Authors · 2025-11-27
> **Further clarification on the differences from Gaussian Garments**
>
> >**Q4. Discussion of Gaussian Garments. (Further clarification)**
>
> Once again, we sincerely appreciate your insightful comment. We realized that the explanation provided in our first response might have been overly brief. To further clarify how our method differs from Gaussian Garments, we conducted an extensive literature review.
>
> We found that the existing work **EUNet** [1] shares a conceptual similarity with ours: **EUNet learns the constitutive law of the material from the motion trajectory of a single square cloth patch, and then uses the learned energy model to drive 3D animations of various garments** (e.g., T-shirts, dresses, skirts).
> However, unlike EUNet, which pre-trains a GNN on node trajectories, our goal is to learn cloth dynamics purely from visual observations, in an unsupervised manner and under unknown physical conditions. We have cited this related work in the **Introduction** section. The corresponding modification is highlighted in light green.
>
>
> [1] Learning 3D Garment Animation from Trajectories of A Piece of Cloth, NIPS 2024

---

> ### Author Response · Authors · 2025-11-29
> **Thank you for reviewing our paper**
>
> Dear Reviewer kW74,
>
> We truly appreciate the time and effort you devoted to reviewing our paper. Your comments are helpful for improving the quality of our work. We apologize that this note comes later than intended. We had been waiting for the opportunity to respond after you completed any revisions or comments on the initial review. Unfortunately, due to an unexpected incident beyond the control of either party, further direct exchanges were no longer possible. We therefore take this opportunity to express our sincere thanks, albeit belatedly.
>
> Due to the temporary policy introduced after the recent incident at ICLR, we are no longer able to continue the discussion in this review cycle. We hope that our rebuttal has clarified all of your previous questions and further strengthened your confidence in our work.
>
> Best regards,
>
> The Authors

---

### Official Review · Reviewer_bT4z · 2025-11-01

**Soundness:** 3
**Presentation:** 3
**Contribution:** 4
**Rating:** 6
**Confidence:** 5

**Summary:**

The authors in this paper presents a method for cloth dynamics grounding. Cloth dynamics are learned from visual observations under unknown conditions without physical supervision.

**Strengths:**

- The use of  Spatial Mapping Gaussian Splatting to establish a mapping between the 2D pixel space and 3D space is interesting. SMGS handles large deformations and severe self-occlusion by using both relative and absolute positions of the Gaussian components. This design ensures an accurate mapping between the 2D and 3D spaces during rendering.

**Weaknesses:**

- The visual results are shown under wind force, it would have been interesting to see cloth dynamics under various type snd source of forces e.g. objects colliding with cloth. How to model them inside the current framework.
- A detailed analysis on cloth-cloth collision, cloth-object collision is missing.
- Do add following relevant references under neural garment simulator GarSim: Particle Based Neural Garment Simulator WACV 2023 and GenSim: Unsupervised Generic Garment Simulator CVPR 2023

**Questions:**

refer to the weakness section

---

> ### Author Response · Authors · 2025-11-21
> **Rebuttal by Authors**
>
> We sincerely thank you for the positive feedback along with constructive comments, which are very helpful in improving our paper. We have posted the point-to-point reply to each question/comment raised by you. We have updated the paper according to reviews' suggestion, and all revisions related to your comments are highlighted in **light yellow**. The visualization videos referenced in the rebuttal are available at ***https://anonymous.4open.science/r/ICLR_rebuttal_video***. Please do feel free to let us know if you have any further questions or guidance.😊
>
> >**Q1. How to model various type of forces inside the current framework?**
>
> **Reply:** Thanks for your insightful comment！To model various types of forces into our framework, we categorize them into two groups: **environmental forces** and **inter-object forces**. Our experiments show that CloDS effectively models environmental forces (e.g. gravity, wind). To further account for inter-object forces, we follow VGPL [1] by assigning each object a **rigidity attribute**, which constrains the relative motion among its nodes. This additional property enables our framework to model interactions among objects with different material characteristics (Related experiments are provided in **our response to Q2**). The relevant clarification is now included in Section 5.7.
>
> >**Q2. Cloth-object collision, cloth-cloth collision is missing.**
>
> **Reply:** This comment has been extremely helpful in improving the quality of our paper！
>
> - **Cloth-object collision：** We construct a corresponding dataset in which a piece of cloth falls under gravity and interacts with a moving rigid sphere. The visualization results of CloDS on this dataset are presented in **Figure 9b** of the revised paper. Video is available in Part 2 at ***https://anonymous.4open.science/r/ICLR_rebuttal_video***. The quantitative results are presented in **Table A**. We can observe CloDS successfully learns the underlying object–cloth collision dynamics, demonstrating strong generalization to multi-object scenarios. We have updated the corresponding description in Section 5.7 of the paper, with the revisions highlighted in light yellow.
>
>     **Table A.** Average RMSE between predicted mesh nodes and ground truth in Cloth-object collision dataset.
>
>     |Model|RMSE|
>     |:-:|:-:|
>     |**CLODS\*\*(ours)**|0.133 $\pm$ 0.024|
>
>
>
>
> - **Cloth-cloth collision：** To further demonstrate the effectiveness of CloDS in modeling cloth–cloth collisions, and following the suggestions from you and reviewer FVhY, we construct a dynamic real-pants dataset [2]. Collisions between the two pant legs naturally serve as instances of cloth–cloth interaction. The visualization results of CloDS on this dataset are shown in **Figure 9c**, and the corresponding videos are provided in Part 3 at ***https://anonymous.4open.science/r/ICLR_rebuttal_video***. The quantitative results are presented in **Table B**.
>
>     **Table B.** Average RMSE between predicted mesh nodes and ground truth in real pants dataset.
>
>     |Model|RMSE|
>     |:-:|:-:|
>     |**CLODS\*\*(ours)**|0.065 $\pm$ 0.002|
>
> >**Q3: There is a lack of relevant references.**
>
> **Replay:** Great suggestion！We have added the relevant citations in the Related Work section, and the revisions are highlighted in light yellow.
>
>
> **Concluding remark:** We sincerely thank you for putting forward excellent comments. We hope the above responses are helpful to clarify your questions. We look forward to addressing any additional questions. Your consideration of improving the rating of our paper will be much appreciated!
>
> **References:**
>
> [1] Visual grounding of learned physical models, ICML 2020
>
> [2] Deep Fashion3D: A Dataset and Benchmark for 3D Garment Reconstruction from Single Images, ECCV 2020

---

> ### Author Response · Authors · 2025-11-26
> **Kindly request your feedback before the end of the discussion period**
>
> Dear Reviewer bT4z:
>
> As the author-reviewer discussion period is soon ending, we would appreciate it if you could review our responses and provide your feedback at your earliest convenience. If there are any further questions or comments, we will do our best to address them before the discussion period ends.
>
> Thank you very much for your time and efforts! 😊
>
> Sincerely,
>
> The Authors

---

> > ### Comment · Reviewer_bT4z · 2025-11-28
> > **Post Rebuttal**
> >
> > Thanks for the detailed response. I am increasing my final score to 7.

---

> > > ### Author Response · Authors · 2025-11-28
> > > **Thank you for raising the score**
> > >
> > > Dear Reviewer bT4z,
> > >
> > > We greatly appreciate your positive feedback and recognition of the contribution of our paper. The discussion with you has been quite productive and fruitful. Your comments and questions have been extremely helpful in improving the quality of our paper. We have incorporated the corresponding revisions into the manuscript, and the updates are highlighted in light yellow in the rebuttal stage.
> > >
> > > Thank you very much for your consideration of raising the score. BTW, as far as we understand, scores above 6 only include 8 (rather than 7).
> > >
> > > Best regards,
> > >
> > > The Authors

---

### Author Response · Authors · 2025-11-21
**Global Response**

Dear Reviewers:

We would like to thank you for your constructive comments, which are extremely helpful for improving our paper. The baseline, dataset, and writing suggestions you provided have greatly strengthened our work. We have provided a point-by-point response to each question and comment raised by you. In addition, we have conducted new experiments and made the corresponding revisions to the manuscript based on your suggestions.😊

- **Reviewer bT4z**: the corresponding revisions are highlighted in **light yellow**.
- **Reviewer kW74**: the corresponding revisions are highlighted in **light green**.
- **Reviewer FVhY**: the corresponding revisions are highlighted in **light blue**.


The videos associated with these experiments are available at: ***https://anonymous.4open.science/r/ICLR_rebuttal_video***. Specifically, the additional experiments we conducted are as follows:

- **Reviewer FVhY**：Comparison between cloth-adapted geometry-aware approaches and CloDS (***Section 5.6***).
- **Reviewer bT4z**：Cloth–object collision experiments. We constructed a Cloth–Object Collision dataset, in which a piece of cloth falls under gravity and interacts with a moving rigid sphere, and demonstrated the robustness of CloDS on this dataset (***Section 5.7***).
- **Reviewer bT4z、kW74、FVhY**：Real-world garment experiments. We construct a Real-Garment training and test dataset by performing physics-based simulations on high-quality garment meshes from the DeepFashion3D V2 dataset, and we demonstrate the robustness of CloDS on this dataset (***Section 5.7***).
- **Reviewer FVhY**：Visualization under different noise conditions. We provide visual results across varying noise types. (***Appendix Section E***).
- **Reviewer FVhY**：Visualization of the ground-truth mesh, the extracted mesh, the predicted mesh, and the mesh extracted without mesh-connectivity constraints (***Appendix Sections H.3 and H.4***).

In addition, we have made the following major revisions to the manuscript:

- **Reviewer bT4z、kW74、FVhY**：Add relevant references (***Section 2***).
- **Reviewer kW74**：Added a more concise method description and additional explanatory figure (***Sections 1 and 4***).
- **Reviewer kW74**：Corrected a typo in the equation（***section 4.3***）.
- **Reviewer kW74**：Include mean ± std in the result tables.
- **Reviewer kW74**：Added inference time (***Appendix Section D.6***).
- **Reviewer FVhY**：Added explanation to the extraction of the initial mesh (***Appendix Section H.9***).

Please do feel free to let us know if you have any further questions.


Thank you very much.

Best regards,

The Authors of the Paper

---

### Author Response · Authors · 2025-12-01
**Summary of Our Response and Revisions during the Rebuttal Period**

Dear Area Chairs,

Thank you very much for your time and effort in evaluating our paper and reviewing the discussions with the reviewers! To facilitate your assessment and save your time, we summarize our rebuttal into three parts: **(1) reviewer feedback**, **(2) our contributions**, and **(3) Major responses**.


---
**1. Reviewer Feedback**
---
During the rebuttal period, our responses and revisions were **positively acknowledged by reviewers**:

- **``Reviewer bT4z``** : After our rebuttal, **``reviewer bT4z``** further expressed appreciation for our work and **raised their score**: "*Thanks for the detailed response. I am increasing my final score to 7.*" **(6→7) or (6→8)?**
- **``Reviewer kW74``** : We regret that **``reviewer kW74``** were unable to see our detailed responses and revisions, and we were unable to engage in further discussion with **``reviewer kW74``**. But we remain deeply grateful for the valuable suggestions reviewer provided. **(4, no further discussion)**
- **``Reviewer FVhY``** : After our rebuttal, **``reviewer FVhY``** indicated that their major concerns had been addressed and **raised their score**: "*The response has addressed my major concerns and I will raise the score for weak accept.*" **(4→6)**


---
**2. Our Contributions**
---

We introduce Cloth Dynamics Grounding (CDG), a novel scenario that involves unsupervised learning of cloth dynamics from multi-view visual observations. We further propose Cloth Dynamics Splatting (CloDS), an **unsupervised** dynamic learning framework designed **in unknown condition** for CDG.

We reatly encouraged that the reviewers acknowledged the contributions of our work, which we briefly summarize below:

**Novelty:** All Reviewers highlighted that CloDS providing an unsupervised, visual-only framework for Cloth Dynamics Grounding (**``Reviewers bT4z, kW74, FVhY``**) and the 2D–3D mapping used in this work is particularly interesting (**``Reviewers bT4z``**).

**Novel Problem & Setting:** Reviewers noted that our work clearly defines a new and challenging problem (**``Reviewer FVhY``**) and emphasized that CloDS can to learn cloth dynamics without direct physical supervision (**``Reviewers FVhY, bT4z``**).

**Strong Performance:** Reviewers observed that CloDS achieves performance close to fully mesh-supervised methods.(**``Reviewers kW74, FVhY``**)



---
**3. Major Responses**
---

During the rebuttal stage, reviewers provided highly insightful comments and questions, which substantially strengthened our work. In response, we have accordingly revised the manuscript. For your convenience, we summarize the major response below:

**3.1 Additional Experiments**
- Added Cloth–object collision experiments. We constructed a Cloth–Object Collision dataset, in which a piece of cloth falls under gravity and interacts with a moving rigid sphere, and demonstrated the robustness of CloDS on this dataset (**``Reviewer bT4z``**).
- Added Comparison between cloth-adapted geometry-aware approaches and CloDS (**``Reviewer FVhY``**).
- Added Real-world garment experiments. We construct a Real-Garment training and test dataset by performing physics-based simulations on high-quality garment meshes from the DeepFashion3D V2 dataset, and we demonstrate the robustness of CloDS on this dataset (**``Reviewer bT4z、kW74、FVhY``**).

**3.2 Improved Visualization**
- Added visualization under different noise conditions. (**``Reviewer FVhY``**).
- Added visualization of the ground-truth mesh, the extracted mesh, the predicted mesh, and the mesh extracted without mesh-connectivity constraints (**``Reviewer FVhY``**).

**3.3 Improved writing**
- Added relevant references (**``Reviewer bT4z、kW74、FVhY``**).
- Corrected a typo in the equation (**``Reviewer kW74``**).
- Added a more concise method description and additional explanatory figure (**``Reviewer kW74``**).
- Added inference time (**``Reviewer kW74``**).
- Include mean ± std in the result tables (**``Reviewer kW74``**).
- Added explanation to the extraction of the initial mesh (**``Reviewer FVhY``**).


**3.4 Clarification**
- Relation to Gaussian Garments: We clarified that CloDS addresses an upstream problem relative to Gaussian Garments, and after an literature review, we further elaborated on the relationship between the two methods based on existing work (**``Reviewer kW74``**).
- Lighting effects: We clarified that even under complex lighting conditions, CloDS is still able to effectively learn cloth dynamics, and this clarification was acknowledged and appreciated by the reviewer. (**``Reviewer FVhY``**).

---

**Concluding remark:** Once again, we would like to thank the AC, SAC, PC, and all reviewers for your valuable time and effort. We sincerely wish you all the best in your future work!

Best Regards,

Authors

---

### Meta-Review · Area_Chair_mg5z · 2026-01-04

**Summary:**

This paper receives a 6 and two 4 in ratings. After the rebuttal and discussion period, 6 is increased to 7, and one 4 is increased to 6. Main concerns are: 1) lack of evaluation on more challenging cases (cloth-object collision, cloth-cloth collision, DeepFashion3D); 2) missing report of inference time; 3) missing references; 4) unclear details.

During rebuttal and discussion period, authors have provided responses to address all of these concerns.

The decision is accept, and the authors are required to include suggested experiments and clarifications.

**Reviewer Concerns:**

All concerns are addressed.

**Reviewer Scores:**

2 reviewers increased their scores. And remaining reviewer would change his/her score as well.

---

### Decision · Program_Chairs · 2026-01-26

Accept (Poster)